# Unsupervised Open-Set Task Adaptation Using a Vision-Language Foundation Model

## Abstract

Human-labeled data is essential for developing deep learning models that can attain human-level recognition. However, the cost associated with annotation presents a substantial practical challenge when deploying these models in real-world applications. Recently, vision-language models like CLIP have shown remarkable zero-shot learning abilities due to vision-language pre-training. While these models can adapt to various tasks without task-specific human-labeled data, fine-tuning them with such data, though beneficial, can be impractical in cost-sensitive situations. In this paper, we propose an alternative method that harnesses vast amounts of open-set unlabeled data from the wild to establish a robust image classification model suitable for real-world scenarios. Our proposed algorithm, Unsupervised Open-Set Task Adaptation (UOTA), offers a straightforward and practical solution, fully capitalizing on the pre-trained CLIP model to enhance its performance by exhaustively utilizing open-set unlabeled data. We substantiate the effectiveness of our contributions through comprehensive experiments conducted on open-set domain adaptation (OSDA) benchmarks that are relevant to our framework. Remarkably, without leveraging any source domain model or labeled source data, our method substantially enhances CLIP's classification performance and attains state-of-the-art results on these benchmarks.

## 1 Introduction

Large amounts of human-annotated data are generally required to train high-performance deep neural networks. However, collecting such data is costly, posing a challenge for real-world applications. Solutions utilizing unlabeled data (Devlin et al., 2019; Brown et al., 2020; He et al., 2022; Chen et al., 2020; He et al., 2020) have been proposed, but human-labeled data is still required for task-specific learning stages (i.e., task adaptation, fine-tuning, and transfer learning).

Recent studies have proposed a new learning paradigm (Radford et al., 2021; Gao et al., 2021; Jia et al., 2021b; Zhou et al., 2022) that achieves *zero-shot capabilities* by learning transferable representations through vast amounts of image and text pairs, although task-specific human-labeled data is needed to improve downstream performance (Radford et al., 2021; Zhou et al., 2022; Gao et al., 2021). However, to the best of our knowledge, no previous work in the literature has explored real-world scenarios where transfer performance can be enhanced solely by utilizing open-set unlabeled data, including both in-distribution (ID, task-relevant) and out-of-distribution (OOD, task-irrelevant) data.

To address this problem, we begin by considering a scenario in real-world situations where only unlabeled data from a specific origin (*e.g.*, a camera at a specific location) is available. This will be the target domain of our zero-shot model based on CLIP (Radford et al., 2021) for a given downstream task, and we assume all data from it shares some characteristics, such as style and texture. We then assume the realistic, *open-set* setting (Scheirer et al., 2012; Bendale & Boult, 2016; Kong & Ramanan, 2021; Vaze et al., 2022), which does not impose any constraints on the data, where data can be randomly collected from a particular origin and may contain content related to known (i.e., in-distribution; ID) or unknown (i.e., out-of-distribution; OOD) classes.

To improve the transfer performance of CLIP (Radford et al., 2021) using open-set unlabeled data, we propose Unsupervised Open-Set Task Adaptation (UOTA), a simple and practical algorithm that operates within a unified framework based on CLIP as shown in Figure 1.

Our method consists of three objectives: (1) a self-training objective and (2) a OOD training objective based on curriculum learning (Bengio et al., 2009; Zhang et al., 2017; Li et al., 2017; Huang et al., 2020; Zhou et al., 2020; Zhang et al., 2021a;b), where class-wise thresholds for detecting unknown class data and classifying known class data are gradually adjusted according to the training status, and (3) a contrastive objective (Sohn, 2016; van den Oord et al., 2018) to push data with unknown classes away from the space of data with known classes and learn a more discriminative representation space for OOD detection.

Leveraging only open-set unlabeled data, our approach enhances pre-trained CLIP (Radford et al., 2021) to implicitly acquire significantly improved ability to perform OOD detection as well as ID classification during the training process without additional explicit methods (*i.e.*,

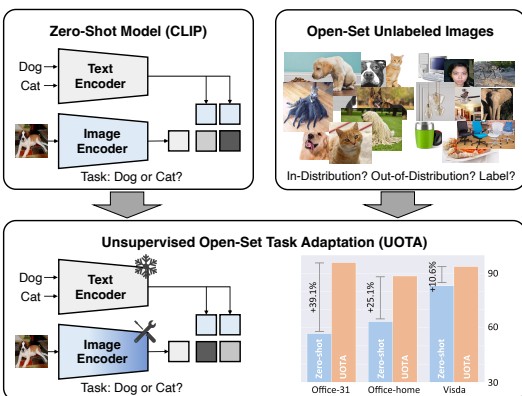

Figure 1: **An overview of our proposed method.** The goal of Unsupervised Open-Set Task Adaptation (UOTA) is to enhance the transfer performance of the zero-shot model (CLIP) on a downstream target task by leveraging open-set unlabeled data in the wild.

Generating candidate unknown classes (Esmaeilpour et al., 2022)) to detect OOD samples. Furthermore, our proposed method is computationally efficient, as it only updates a lightweight adapter inserted in the image encoder while freezing the rest of the model. We validate the effectiveness of UOTA by comparing it with pre-trained CLIP (denoted as "Zero-shot") and methods that can perform open-set domain adaptation (OSDA) or source-free open-set domain adaptation (SF-OSDA) using standard OSDA benchmarks. Note that, as explained in Table 1, UOTA does not use labeled source data or a model trained on labeled source data, which are for OSDA and SF-OSDA, respectively. Despite this, UOTA significantly outperforms models that can perform OSDA and SF-OSDA, while remarkably improving pre-trained CLIP.

In summary, our main contribution is proposing *a simple yet effective method to significantly enhance the classification task performance of a zero-shot model (e.g., CLIP) in a realistic open-set unlabeled data setting* where only unlabeled data collected from a specific origin is provided, and each sample can be either task-relevant (known class, ID) or irrelevant (unknown class, OOD).

## 2 RELATED WORK

**Multimodal zero-shot model**    CLIP (Radford et al., 2021) is a new method for open-vocabulary zero-shot image classification using natural language supervision on large datasets. ALIGN (Jia et al., 2021a) and SLIP (Mu et al., 2022) improved CLIP by aligning vision-language representations in a latent space and using self-supervision, respectively. Some recent works have attempted to adapt CLIP to downstream tasks using labeled data (Zhou et al., 2022; Gao et al., 2021) or unsupervised fine-tuning (Li et al., 2022). Moreover, (Ming et al., 2022; Esmaeilpour et al., 2022) conduct OOD detection using pre-trained CLIP, but Ming et al. (2022) is a training-free method that does not improve the OOD detection capability that pre-trained CLIP originally possesses, and Esmaeilpour et al. (2022) explicitly generates candidate OOD class prompts. We propose, for the first time in the literature, a novel method that simultaneously improves the OOD detection and image classification capabilities of CLIP by utilizing only open-set unlabeled data and without using any explicit methods for OOD detection such as generating candidate class prompts (Esmaeilpour et al., 2022).

**Open-set domain adaptation**    In real-world scenarios, the set of classes in the target distribution may expand to include *unknown* classes, which leads to the field of open-set domain adaptation (OSDA) (Saito et al., 2018; Liu et al., 2019). Conventional OSDA methods have focused on aligning the features of known classes in the source and target domains through domain adversarial learning (Saito et al., 2018; Liu et al., 2019). Some methods proposed more advanced approach by learning intrinsic target structures through self-supervised learning (Li et al., 2021; Saito et al., 2020). While general OSDA methods allow access to source domain data during the adaptation stages, recently proposed source-free OSDA (SF-OSDA) (Yang et al., 2022; Liang et al., 2020) methods

Table 1: **Comparison of settings.** In Unsupervised Domain Adaptation (UDA), models are initially trained on the labeled source domain data ($D_s$). During the adaptation stage, both the labeled source domain data ($D_s$) and the unlabeled target domain data ($D_t$) are employed to adapt the model to the target domain. It assumes a *closed-set setting* where the source domain classes ($C_s$) are equal to the target domain classes ($C_t$), so an unknown class does not exist in $D_t$. OSDA follows the UDA framework, but the only difference is that $C_s$ is a subset of $C_t$, thereby suggesting an *open-set setting* where unknown classes are present in $D_t$. SF-OSDA basically follows OSDA, but it does not utilize $D_s$ during the adaptation stage. However, it still uses a model trained on $D_s$. ODAwVL ([Yu et al., 2023](#)) principally adheres to SF-OSDA, employing a model trained on $D_s$ and not using $D_s$ during the adaptation. However, they additionally use guidance from pre-trained CLIP during the adaptation. In contrast to all previous methods, UOTA assumes a more restrictive *open-set unlabeled data setting*, which neither uses a model trained on $D_s$ nor $D_s$ itself. It only uses $D_t$ for adaptation.

| Setting Comparison | UDA | OSDA | SF-OSDA | ODAwVL | Ours |
|---|---|---|---|---|---|
| Pre-training Dataset / Model | ImageNet | ImageNet | ImageNet | ImageNet+CLIP | CLIP |
| Unlabeled Target Data during Adaptation | ✔ | ✔ | ✔ | ✔ | ✔ |
| Unknown Classes Allowed (Open-Set Setting) | - | ✔ | ✔ | ✔ | ✔ |
| No Source Data during Adaptation (Source-Free) | - | - | ✔ | ✔ | ✔ |
| No Pre-training on Source Data | - | - | - | - | ✔ |

utilize a model trained on the source domain but do not use source data during the adaptation stage. ODAwVL ([Yu et al., 2023](#)) utilized the guidance of CLIP for training the SF-OSDA model, but still used the model trained on labeled source domain data. In this paper, we propose a more restrictive setting than previous SF-OSDA, where the model uses neither source domain data nor a model trained on source data as shown in Table 1.

## 3 Method

UOTA fully exploits the pre-trained CLIP model that has a dual-stream architecture with a text encoder $\mathcal{T}_\phi$ and an image encoder $\mathcal{I}_\theta$, where $\phi$ and $\theta$ are the pre-trained parameters. For a given downstream task $\tau$ with a class set $Y_\tau = \{y_i\}_{i=1}^{K_\tau}$, where $K_\tau$ denotes the number of classes to be classified, we first complete a set of class embeddings $\mathcal{C}_\tau = \{\mathcal{T}_\phi(p_i)\}_{i=1}^{K_\tau}$ by using natural language prompting $p_i = $ "a photo of a {class name of $y_i$}". When image data $x$ is given, the corresponding embedding $\mathcal{I}_\theta(x)$ is compared with the class embeddings by measuring the cosine similarity, and then we compute the task-wise classification probability as:

$$p(y = y_i|x; \phi, \theta) = \frac{e^{\alpha \cdot \text{Sim}(\mathcal{I}_\theta(x), \mathcal{T}_\phi(p_i))}}{\sum_{j=1}^{N} e^{\alpha \cdot \text{Sim}(\mathcal{I}_\theta(x), \mathcal{T}_\phi(p_j))}}, \tag{1}$$

where $\alpha$ is a learnable scaling factor (*i.e.*, temperature) and $\text{Sim}(\cdot, \cdot)$ denotes cosine similarity between two vectors. The overall architecture is shown in Figure 2.

### 3.1 Self-training with open-set unlabeled data

We compute and utilize the maximum value $\max_i p(y = y_i|x, \phi, \theta)$ of the predicted probability ($= s_{max}$, maximum probability score), described in Equation 1, for detecting OOD samples in the dataset $\mathcal{D}$ by using a CLIP model. If an image $x$ belongs to the in-distribution (ID), the similarity to one of the known class embeddings $\mathcal{C}_\tau$ will be high and result in a high maximum probability score. Conversely, if it belongs to an out-of-distribution (OOD) class, there will be no matching known class embedding, resulting in a low maximum probability score. We can confidently identify an image as ID if its maximum probability score is above a certain threshold for ID (*i.e.*, $\max_i p(y = y_i|x, \phi, \theta) \geq \delta_{\text{in}}$) and as OOD if $1 - s_{max}$ is above another threshold for OOD (*i.e.*, $1 - \max_i p(y = y_i|x, \phi, \theta) \geq \delta_{\text{out}}$).

Our approach is novel in that, for the first time in the literature, it simultaneously adjusts both $\delta_{\text{in}}$ and $\delta_{\text{out}}$ based on the model's learning status for each class. This method aligns well with the curriculum learning strategy ([Bengio et al., 2009](#)) and ensures that the model adaptively focuses on the confident images, improving its ability to detect OOD samples as well as precisely classify the ID classes.

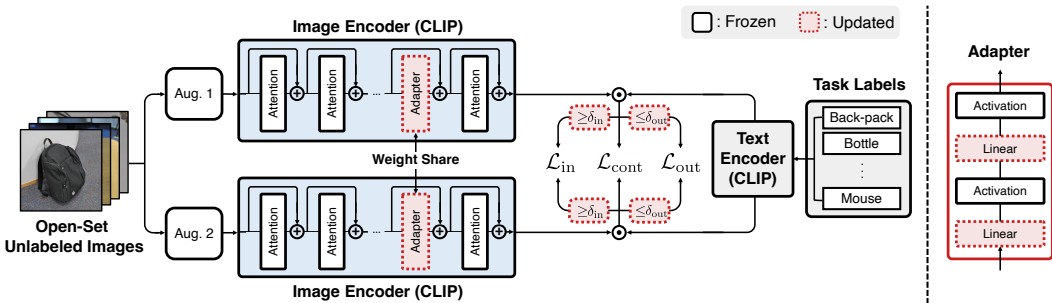

Figure 2: **Left: Implementation details of Unsupervised Open-Set Task Adaptation (UOTA).** Our framework improves the adaptation performance of CLIP by leveraging both in-distribution (ID) and out-of-distribution (OOD) samples. **Right: Architecture of the adapter module.** The adapter in our framework comprises two linear layers and two activation layers. During the training process, we update only this lightweight adapter, enabling computationally efficient training.

**Gradual adjustment of class-wise thresholds**    Our adjustable class-wise thresholds for both ID and OOD are defined by scaling the fixed thresholds $\delta_{\text{in}}$ and $\delta_{\text{out}}$ as:

$$\delta_{\text{in}}(y_i) = \beta_{\text{in}}(y_i) \cdot \delta_{\text{in}} \quad \text{and} \quad \delta_{\text{out}}(y_i) = \beta_{\text{out}}(y_i) \cdot \delta_{\text{out}}, \tag{2}$$

where the class-wise scaling factors $\beta_{\text{in}}(y_i)$ and $\beta_{\text{out}}(y_i)$ are computed in the same manner and updated regularly (*i.e.*, at each epoch). For example, we update the class-wise scaling factor $\beta_{\text{in}}(y_i)$ for ID as:

$$\beta_{\text{in}}(y_i) = \frac{n_{\text{in}}(y_i) + \gamma \cdot \max_j n_{\text{in}}(y_j)}{(1 + \gamma) \cdot \max_j n_{\text{in}}(y_j)}, \tag{3}$$

where $n_{\text{in}}(y_i)$ denotes the number of samples in the dataset $\mathcal{D}$ whose classes are predicted as $y_i$ while presenting maximum probability scores higher than $\delta_{\text{in}}$. Here, $\gamma$ denotes a smoothness factor to reduce the variability of scaling factors between classes. We experimentally found there is a negligible performance variation with slight changes in $\gamma$. We also update the class-wise scaling factor $\beta_{\text{out}}(y_i)$ for OOD using the same equation, where $n_{\text{out}}(y_i)$ are collected similarly but differently, with $1 - s_{max}$ higher than $\delta_{\text{out}}$. Additionally, we empirically found that updating $n_{\text{in}}$ and $n_{\text{out}}$ as a moving average results in a more stable learning, further improving the performance of UOTA. This flexible modification of $\delta_{\text{in}}$ and $\delta_{\text{out}}$ based on the model's training status allows the model to implicitly and accurately separate OOD samples from ID ones without additional explicit methods (*i.e.*, Utilizing candidate unknown classes (Esmaeilpour et al., 2022)). This also prevents a considerable amount of data from being discarded due to imprecise, fixed thresholding.

**Self-training with in-distribution data**    We first obtain pseudo-labels using two randomly augmented views of $x$, denoted as $\mathcal{A}_1(x)$ and $\mathcal{A}_2(x)$. Since we deal with open-set unlabeled data, some pseudo-labels may be generated from OOD samples, but utilizing such samples for pseudo-labeling may negatively affect the model's training. Therefore, we aim to employ only the images confidently predicted as ID for generating pseudo-labels. Concretely, for each image $x$, we formulate the sample-level self-training loss $\mathcal{L}'_{\text{in}}$ as:

$$\mathcal{L}'_{\text{in}}(p_1(x), p_2(x)) = \mathbb{1}_{[\max p_1(x) \geq \delta_{\text{in}}(\hat{p}_1(x))]} \mathcal{L}_{\text{ce}}(\hat{p}_1(x), p_2(x)), \tag{4}$$

where $p_1(x)$ and $p_2(x)$ are the predicted probabilities $p(y|\mathcal{A}_1(x), \phi, \theta)$ and $p(y|\mathcal{A}_2(x), \phi, \theta)$, respectively. In this loss, $p_1(x)$ is treated as the target distribution of $p_2(x)$, and thus we use the cross entropy loss $\mathcal{L}_{\text{ce}}$ by making a hard pseudo-label $\hat{p}_1(x)$ from $p_1(x)$. By using our class-wise thresholds $\delta_{\text{in}}(\cdot)$, we adaptively mask out this loss if the image $x$ is not confidently predicted as ID. With this sample-level self-training loss $\mathcal{L}'_{\text{in}}$, we formulate the overall loss related to ID data as:

$$\mathcal{L}_{\text{in}} = \frac{1}{2|\mathcal{B}|} \sum_{x \in \mathcal{B}} \mathcal{L}'_{\text{in}}(p_1(x), p_2(x)) + \mathcal{L}'_{\text{in}}(p_2(x), p_1(x)), \tag{5}$$

where we symmetrically compute the loss with two differently predicted probabilities $p_1(x)$ and $p_2(x)$ for each $x$ and average over a mini-batch $\mathcal{B}$ sampled from the dataset $\mathcal{D}$.

**Utilizing out-of-distribution data**  Since OOD samples do not have matching classes in $Y_\tau$, we assume that forcing them to move away from the corresponding class embedding space can improve the model's capability to detect OOD data. Therefore, we propose to reduce the maximum probability scores of OOD samples during task adaptation, and this can be treated as regularization that makes the model's prediction for OOD data more unconfident (*i.e.*, makes the prediction probability near uniform). For each sample $x$, we define the sample-level negative learning loss $\mathcal{L}'_{\text{out}}$ as:

$$\mathcal{L}'_{\text{out}}(p_1(x), p_2(x)) = \mathbb{1}_{[1-\max p_1(x) \geq \delta_{\text{out}}(\hat{p}_1(x))]} \mathcal{L}_{\text{ce}}(\hat{p}_1(x), 1 - p_2(x)). \tag{6}$$

Using this loss, we aim to maximize the cross-entropy between $p_1(x)$ and $p_2(x)$ by simply treating $p_1(x)$ as the target distribution of $1-p_2(x)$. To properly apply this loss to only OOD samples, similar to Equation 4, we apply adaptive masking to each image $x$ using class-wise thresholds $\delta_{\text{out}}(\cdot)$ that selectively choose samples with unconfident predictions (*i.e.*, lower maximum probability scores). Based on this sample-level loss $\mathcal{L}'_{\text{out}}$, we formulate the overall loss related to OOD data as:

$$\mathcal{L}_{\text{out}} = \frac{1}{2|\mathcal{B}|} \sum_{x \in \mathcal{B}} \mathcal{L}'_{\text{out}}(p_1(x), p_2(x)) + \mathcal{L}'_{\text{out}}(p_2(x), p_1(x)). \tag{7}$$

Our proposed loss related to OOD data aligns well with negative learning (Kim et al., 2019), yet ours possesses distinct and novel characteristics that markedly differentiate it from (Kim et al., 2019). Kim et al. (2019) uses fixed thresholds to filter noisy data, while we propose a new method that updates the class-wise OOD thresholds considering training status to improve OOD detection accuracy as training progresses. Also, Kim et al. (2019) randomly chooses a complementary label for the noisy data from given labels, while we use a label of differently augmented view as the complementary label for OOD data to align with our self-training scheme for ID data. Moreover, while Kim et al. (2019) updates whole network parameters several times in a single iteration using sequential losses and has a complicated multi-stage training process, we jointly minimize all losses to update a lightweight adapter only once in a single iteration for computational efficiency and employ a simple end-to-end training.

**Contrastive loss as an additional regularizer**  Recent studies (Winkens et al., 2020; Tack et al., 2020) have demonstrated that the use of contrastive loss can improve OOD detection by enriching the representation space. In that way, we adopt the contrastive loss $\mathcal{L}_{\text{cont}}$ proposed in SimCLR (Chen et al., 2020) to all given data, regardless of whether it is ID or OOD data, and use it as a regularizer to enhance not only OOD detection but also the adaptation performance.

Based on the objectives described thus far, we fully utilize both ID and OOD samples in the open-set unlabeled dataset to update the lightweight adapter in the image encoder $\mathcal{I}_\theta$ of the CLIP during task adaptation, and the overall loss is:

$$\mathcal{L} = \mathcal{L}_{\text{in}} + \mathcal{L}_{\text{out}} + \omega \cdot \mathcal{L}_{\text{cont}}, \tag{8}$$

where $\omega$ is used as a balancing hyper-parameter. After the task adaptation is finalized by optimizing the model with this overall loss, we use a fixed threshold $\delta_{\text{ood}}$ at test time to detect OOD samples by simply comparing it with the maximum probability scores.

## 4  EXPERIMENTS

In this section, we provide a detailed overview of our experimental settings, which include the datasets, baselines, and evaluation metrics. We then present a comprehensive quantitative and qualitative analysis of UOTA and compare it with state-of-the-art models on open-set domain adaptation (OSDA) and source-free OSDA (SF-OSDA). Note that UOTA is the first approach to perform task adaptation using only open-set unlabeled data and without any source domain model or data as explained in Table 1. As a result, technically, there are no comparable models or experimental protocols available. Therefore, we compare our approach with the models that can perform OSDA and SF-OSDA using benchmarks utilized in the OSDA. Additionally, we provide a detailed ablation study for each component of UOTA.

Table 2: **Experiment results on Office-31, Office-Home, and VisDA**. We utilize the HOS score (%) as an evaluation metric. As explained in Table 1, OSDA employs both the labeled source data and the target data during the adaptation stage. Source-free OSDA employs models trained on labeled source data but use only target data during the adaptation stage. In contrast with OSDA or source-free OSDA, UOTA only utilizes unlabeled target data and does not use either the source data or the model trained on source data.

| Method | Office-31 | | | | | | | Office-Home | | | | | | | | | | | | | Visda |
|---|---|---|---|---|---|---|---|---|---|---|---|---|---|---|---|---|---|---|---|---|---|
| | W | D | A | D | A | W | Avg. | R | C | A | P | C | A | P | R | A | P | R | C | Avg. | S |
| | A | | W | | D | | | P | | | R | | | C | | | A | | | | R |
| **OSDA** (use labeled source domain data during the adaptation) | | | | | | | | | | | | | | | | | | | | | |
| DANN | 72.6 | 73.7 | 68.1 | 86.7 | 71.5 | 82.5 | 75.9 | 68.4 | 60.9 | 65.2 | 69.8 | 66.7 | 71.0 | 44.6 | 50.9 | 51.2 | 56.3 | 65.4 | 57.6 | 60.7 | - |
| CDAN | 71.0 | 72.7 | 64.9 | 84.3 | 66.8 | 80.5 | 73.4 | 67.6 | 61.7 | 65.1 | 69.7 | 67.1 | 70.7 | 47.2 | 52.7 | 52.9 | 58.6 | 66.0 | 58.2 | 61.4 | - |
| STA | 66.1 | 73.2 | 75.9 | 69.8 | 75.0 | 75.2 | 72.5 | 64.5 | 60.4 | 54.0 | 69.5 | 66.8 | 68.3 | 53.2 | 54.5 | 55.8 | 61.9 | 67.1 | 57.4 | 61.1 | 72.7 |
| OSBP | 73.7 | 75.1 | 82.7 | 97.2 | 82.4 | 91.1 | 83.7 | 72.3 | 64.7 | 65.2 | 73.9 | 70.6 | 72.9 | 53.2 | 54.5 | 55.1 | 63.2 | 66.7 | 64.3 | 64.7 | 69.8 |
| PGL | 70.1 | 69.5 | 74.6 | 76.5 | 72.8 | 72.2 | 72.6 | 52.5 | 36.8 | 45.6 | 41.6 | 45.6 | 55.8 | 46.6 | 0.0 | 29.3 | 47.2 | 11.4 | 10.0 | 35.2 | 74.7 |
| ROS | 77.2 | 77.9 | 82.1 | 96.0 | 82.4 | 99.7 | 85.9 | 75.7 | 65.2 | 69.3 | 74.4 | 68.6 | 76.5 | 56.3 | 60.4 | 60.1 | 60.6 | 68.8 | 58.9 | 66.2 | - |
| DANCE | 70.2 | 65.8 | 66.9 | 80.0 | 70.7 | 84.8 | 73.1 | 44.0 | 45.9 | 49.8 | 41.2 | 30.2 | 39.4 | 55.7 | 48.3 | 53.1 | 54.2 | 27.5 | 40.9 | 44.2 | - |
| DCC | 84.4 | 85.5 | 87.1 | 91.2 | 85.5 | 87.1 | 86.8 | 62.7 | 66.6 | 67.4 | 64.0 | 67.0 | 80.6 | 52.8 | 76.9 | 52.9 | 59.5 | 56.0 | 49.8 | 64.2 | 70.7 |
| OSLPP | 78.7 | 79.3 | 89.0 | 92.3 | 91.5 | 93.6 | 87.4 | 74.4 | 66.9 | 72.8 | 74.0 | 70.4 | 74.3 | 59.3 | 59.0 | 61.0 | 63.6 | 67.2 | 60.9 | 67.0 | - |
| UADAL | 76.5 | 79.7 | 89.1 | 97.8 | 86.0 | 99.5 | 88.1 | 76.8 | 69.5 | 70.8 | 76.9 | 73.4 | 77.4 | 56.6 | 60.6 | 63.2 | 63.0 | 72.1 | 64.2 | 68.7 | 75.3 |
| cUADAL | 75.1 | 80.5 | 90.1 | 98.2 | 87.9 | 99.4 | 88.5 | 76.7 | 68.3 | 71.6 | 76.8 | 72.6 | 77.5 | 54.6 | 59.9 | 63.6 | 62.9 | 72.6 | 65.0 | 68.5 | 75.9 |
| ODAwVL | 91.0 | 91.6 | 92.1 | 93.5 | 93.7 | 95.0 | 92.8 | 79.5 | 77.3 | 76.0 | 82.7 | 82.4 | 83.4 | 76.5 | 76.1 | 76.7 | 82.0 | 82.8 | 81.5 | 79.4 | 80.7 |
| **SF-OSDA** (do not use labeled source domain data during the adaptation but use a source model trained on it) | | | | | | | | | | | | | | | | | | | | | |
| SHOT | 75.9 | 74.0 | 69.1 | 87.2 | 67.2 | 92.7 | 77.7 | 42.3 | 40.2 | 39.8 | 46.2 | 39.1 | 47.0 | 40.8 | 40.1 | 39.5 | 57.7 | 59.9 | 54.6 | 45.6 | 42.6 |
| AaD | 73.9 | 73.0 | 78.3 | 91.2 | 77.7 | 93.5 | 81.3 | 70.1 | 61.4 | 66.9 | 70.6 | 67.8 | 69.9 | 55.9 | 57.5 | 57.6 | 60.1 | 64.6 | 60.5 | 63.6 | 16.0 |
| ODAwVL | 90.8 | 91.9 | 89.3 | 92.6 | 93.3 | 93.8 | 91.9 | 78.9 | 78.6 | 79.3 | 84.6 | 84.8 | 85.5 | 76.5 | 76.8 | 76.5 | 82.2 | 82.1 | 82.3 | 80.7 | 83.81 |
| **Our Setting** (neither use a source domain model nor source domain data) | | | | | | | | | | | | | | | | | | | | | |
| Zero-shot | 48.0 | | 57.0 | | 65.3 | | 56.8 | 57.4 | | | 63.9 | | | 63.1 | | | 69.2 | | | 63.4 | 83.1 |
| **UOTA** | **93.8** | | **94.7** | | **99.3** | | **96.0** | **92.8** | | | **92.2** | | | **85.4** | | | **83.7** | | | **88.5** | **93.7** |
| Zero-shot (Oracle) | 96.8 | | 97.8 | | 98.0 | | 97.5 | 97.0 | | | 98.7 | | | 93.6 | | | 96.3 | | | 96.4 | 94.0 |
| **UOTA (Oracle)** | **97.2** | | **100** | | **100** | | **99.1** | **98.1** | | | **98.9** | | | **95.1** | | | **97.1** | | | **97.3** | **96.1** |

## 4.1 EXPERIMENTAL SETTINGS

We evaluated the performance of UOTA by following the experimental settings of UADAL (Jang et al., 2022) and using a variety of benchmark datasets, including (i) Office-31 (Saenko et al., 2010), (ii) Office-Home (Venkateswara et al., 2017), (iii) VisDA (Peng et al., 2017), and (iv) Domain-Net (Peng et al., 2019). Note that no existing OSDA or SF-OSDA model has conducted experiments on the full DomainNet dataset, and there is no established experimental protocol for it. Therefore, to maintain a similar ratio of ID classes as applied in the other datasets, we set 100 classes out of the total 345 categories as known classes and the remaining as unknown classes.

To quantitatively evaluate the performance of UOTA, we compare it with several existing methods that can perform OSDA, including DANN (Ganin & Lempitsky, 2015), CDAN (Long et al., 2018), STA (Liu et al., 2019), OSBP (Saito et al., 2018), ROS (Bucci et al., 2020), DANCE (Saito et al., 2020), DCC (Li et al., 2021), UADAL (cUADAL) (Jang et al., 2022), and ODAwVL (Yu et al., 2023). Note that some of these models (*e.g.*, CDAN, DANN, *etc.*) are not tailored for OSDA. However, they can still perform OSDA and given that they are frequently used for comparison, we also utilized them. Furthermore, we compare UOTA with state-of-the-art models that can perform SF-OSDA such as SHOT (Liang et al., 2020), AaD (Yang et al., 2022), and ODAwVL (Yu et al., 2023). SHOT and AaD can also perform source-free *closed-set* domain adaptation, but we utilized them only for SF-OSDA. Note that ODAwVL (Yu et al., 2023) is fundamentally an SF-OSDA method. However, by using labeled source domain data during the adaptation stage, it can also conduct OSDA. Therefore, for ODAWVL, we present results for both OSDA and SF-OSDA. We also demonstrate the effectiveness of UOTA by comparing it with a pre-trained CLIP model (Radford et al., 2021), denoted as "Zero-shot". In particular, it is used as our initialization, and the goal of our method is to further improve it.

To effectively evaluate the performance of UOTA, we utilized the HOS metric, which is commonly used as an evaluation criterion by existing OSDA approaches (Bucci et al., 2020; Jang et al., 2022; Yang et al., 2022). The HOS metric is calculated by taking the harmonic mean of OS* and UNK, where OS* represents the mean accuracy over known classes and UNK represents the accuracy of the unknown class. This metric is particularly suitable for evaluating models in OSDA tasks as it considers both known and unknown (ID and OOD) classification capabilities, providing a higher evaluation to models that excel in both. Therefore, we follow the established protocols of OSDA and mainly employ the HOS score as the evaluation metric.

## 4.2 QUANTITATIVE ANALYSIS

Table 2 and 3 show that UOTA outperforms "Zero-shot (pre-trained CLIP)" on all benchmark datasets and all target domains, indicating that UOTA improves the CLIP's capabilities to precisely distinguish OOD samples from ID samples as well as to classify ID samples. The results also show that UOTA consistently exhibits superior performance compared to the models that can perform OSDA or SF-OSDA across all benchmark datasets, despite the more challenging setting assumed for UOTA as shown in Table 1. Note that the key point of our experiments

Table 3: **Experiment results on DomainNet.** We utilize the HOS score (%) as an evaluation metric. For SF-OSDA methods, we average scores over different source domains for each target domain. Our setting does not use either the source data or a model trained on the source data.

| METHOD | DOMAINNET | | | | | | |
|---|---|---|---|---|---|---|---|
| | C | I | P | Q | R | S | AVG. |
| SF-OSDA | | | | | | | |
| SHOT | 43.1 | 23.4 | 34.0 | 12.3 | 39.8 | 36.9 | 31.6 |
| AAD | 49.4 | 24.6 | 30.9 | 15.3 | 51.8 | 42.6 | 35.8 |
| OUR SETTING | | | | | | | |
| ZERO-SHOT | 77.6 | 65.7 | 74.3 | 34.8 | 81.1 | 76.3 | 68.3 |
| **UOTA** | **82.4** | **68.7** | **77.0** | **35.9** | **85.2** | **77.3** | **71.1** |
| ZEROSHOT (ORACLE) | 93.5 | 82.2 | 89.9 | 49.3 | 96.2 | 91.0 | 83.7 |
| **UOTA (ORACLE)** | **94.3** | **84.7** | **91.4** | **58.4** | **96.3** | **91.4** | **86.1** |

is that UOTA showed results that were significantly enhanced compared to "Zero-shot". As explained in Table 1, the pre-trained backbone we used differs from other OSDA and SF-OSDA models. While ODAwVL (Yu et al., 2023) utilizes the pre-trained CLIP, this method simultaneously employs an ImageNet (Deng et al., 2009) pre-trained model, which is additionally trained on the labeled source domain data. (Although it uses an additional model along with pre-trained CLIP, it still shows inferior results compared to UOTA) For other OSDA and SF-OSDA models, experiments were conducted under the best settings they proposed, which means an ImageNet pre-trained backbone was used instead of CLIP. Also, in contrast with OSDA and SF-OSDA, our proposed novel setting does not use the source domain model or data as mentioned in Table 1. Thus, strictly speaking, a direct comparison between UOTA and OSDA (or SF-OSDA) is not appropriate. The results from other methods that can perform OSDA and SF-OSDA serve merely as a reference point, and the core of our experiment is that UOTA significantly improved pre-trained CLIP's capabilities of conducting both OOD detection and ID classification. However, although it is not important for validating the effectiveness of UOTA, we investigated the performance of various methods that can perform OSDA and SF-OSDA (*e.g.*, UADAL, cUADAL, SHOT, AaD, *etc.*) when the CLIP encoder was used as the backbone. Since these models did not provide experimental results with CLIP as the backbone, we carried out test on our own. Because these models are not tailored for use in vision-language models, the results showed they did not converge well, generally achieving less than half the performance of the best settings. Furthermore, concerning the experiments on DomainNet, while ODAwVL did report results for DomainNet, they did not utilize the full dataset. Instead, they selected 126 classes out of the total of 345. Since there are no official codes and it's unclear which specific classes they chose, we excluded ODAWVL from our experiments.

Additionally, to evaluate the effectiveness of the training strategy and determine the maximum performance (*i.e.*, upper-bound) that our method can achieve, we assess the performance of UOTA in an *oracle* setting. The *oracle* setting assumes an experimental environment where accurate separation of ID and OOD classes is provided during training. As one of UOTA's training objectives ($L_{out}$) mainly focuses on learning a discriminative representation for ID and OOD separation, the effect of its loss functions is largely offset in this setting. Nonetheless, UOTA consistently shows performance improvement over "Zero-shot" across all datasets and all target domains.

## 4.3 QUALITATIVE ANALYSIS

In this section, we further demonstrate the effectiveness of UOTA in distinguishing OOD samples from ID ones by visualizing the performance gap between "Zero-shot" and UOTA. We mainly utilize the Office-31 dataset and all of its domains (A, D, and W) as target tasks. First, we compare the maximum probability score distributions obtained from "Zero-shot" and UOTA by visualizing them using histograms. The horizontal axis represents the maximum probability score, while the vertical axis indicates the number of samples. As shown in Figure 3, we observe that "Zero-shot" is unable to clearly distinguish between ID (green) and OOD (red) samples. In contrast, UOTA effectively separates ID and OOD samples by predicting generally low maximum probability scores for OOD samples and high scores for ID samples.

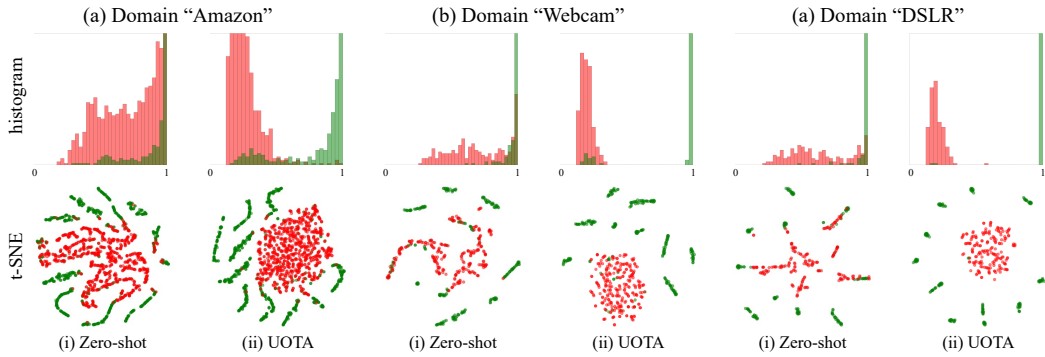

Figure 3: **Histogram and t-SNE visualization on Office-31.** The visualization results for (a) domain A, (b) domain W, and (c) domain D of the Office-31 dataset are shown with histograms and t-SNE plots. Across all domains, UOTA consistently exhibits improved performance over "Zero-shot", with OOD samples (red) appearing more tightly clustered and a clearer separation between ID (green) and OOD samples.

In the next step, we present the t-SNE visualizations of the learned features by "Zero-shot" and UOTA in Figure 3. Each data point in the figure represents the classification probability vector (as described in Equation 1) for each sample. The figure illustrates that the features for OOD samples (red) obtained by "Zero-shot" are not well distinguished from the features for ID samples (green). In contrast, UOTA precisely segregates OOD samples from ID ones. Lastly, we measure the distance between the ID and OOD feature distributions produced by "Zero-shot" and UOTA. For this, we use Proxy $\mathcal{A}$-Distance (PAD) (Ganin et al., 2016) and Maximum Mean Discrepancy (MMD) (Ghifary et al., 2014), and the corresponding results are shown in Figure 4. Higher PAD and MMD values indicate clearer discrimination between ID and OOD feature distributions. Our analysis reveals that UOTA (green) exhibits approximately 50% and 15% higher PAD and MMD, respectively, compared to "Zero-shot (red)". This suggests that UOTA is better able to distinguish between ID and OOD feature distributions.

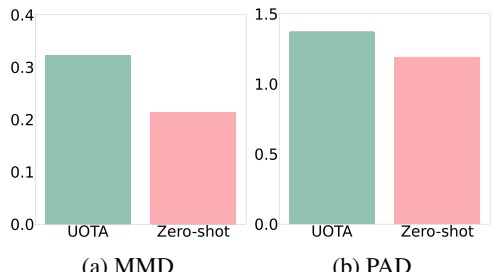

(a) MMD          (b) PAD

Figure 4: **MMD and PAD values between known and unknown feature distributions.** UOTA (green) consistently exhibits a noticeable improvement in both metrics over "Zero-shot" (pre-trained CLIP, red). Each metric value is an average result across all domains of Office-31. This result demonstrates that UOTA more accurately distinguishes between ID and OOD distributions in comparison to "Zero-shot".

### 4.4 ABLATION STUDY

**Effectiveness of the proposed training objectives.** In this section, we conduct an ablation study on the loss components of UOTA using the Office-31 dataset and HOS score. In Table 4, we compare eight different cases: (1) "Zero-shot", (2) "$\mathcal{L}_{\text{in}}$", (3) "$\mathcal{L}_{\text{out}}$", (4) "$\mathcal{L}_{\text{cont}}$", (5) "$\mathcal{L}_{\text{in}} + \mathcal{L}_{\text{cont}}$", (6) "$\mathcal{L}_{\text{out}} + \mathcal{L}_{\text{cont}}$", (7)"$\mathcal{L}_{\text{in}} + \mathcal{L}_{\text{out}}$", and (8) UOTA. When using only "$\mathcal{L}_{in}$", the model diverges and shows a gradual reduction in the HOS score as the inaccurate separation of ID and OOD samples persists during the training process. On the other hand, when only "$\mathcal{L}_{\text{out}}$" is used, the OOD samples gradually separate from the ID samples, and the

Table 4: **Ablation on the proposed training objectives.** The result demonstrates that the performance of the model is maximized when all losses that constitute the training objectives are used together rather than any one of them being omitted.

| METHOD | OFFICE-31 | | | |
|---|---|---|---|---|
| | A | W | D | AVG. |
| ZERO-SHOT | 48.0 | 57.0 | 65.3 | 56.8 |
| $\mathcal{L}_{\text{in}}$ | DIVERGED | | | |
| $\mathcal{L}_{\text{out}}$ | 87.6 | 89.3 | 92.6 | 89.8 |
| $\mathcal{L}_{\text{cont}}$ | 66.9 | 64.2 | 68.1 | 66.4 |
| $\mathcal{L}_{\text{in}} + \mathcal{L}_{\text{cont}}$ | 47.6 | 57.4 | 66.3 | 57.1 |
| $\mathcal{L}_{\text{out}} + \mathcal{L}_{\text{cont}}$ | 89.2 | 89.9 | 92.9 | 90.7 |
| $\mathcal{L}_{\text{in}} + \mathcal{L}_{\text{out}}$ | 92.9 | 94.7 | 94.6 | 94.1 |
| $\mathcal{L}_{\text{in}} + \mathcal{L}_{\text{out}} + \mathcal{L}_{\text{cont}}$ (UOTA) | **93.8** | **94.7** | **99.3** | **96.0** |

model shows fairly good performance for all domains. For the "$\mathcal{L}_{\text{cont}}$" case, the model achieves

better performance than "Zero-shot" through self-supervised learning. However, since it does not have any learning objectives for dividing ID and OOD samples, it does not show satisfying performance. When "$\mathcal{L}_{in}$" and "$\mathcal{L}_{cont}$" are used together, the model still does not accurately detect OOD samples well due to absence of "$\mathcal{L}_{out}$". If "$\mathcal{L}_{out}$" and "$\mathcal{L}_{cont}$" are used together, the HOS score improves for all domains when compared to the case where only "$\mathcal{L}_{out}$" is used. For the "$\mathcal{L}_{in}$ + $\mathcal{L}_{out}$" case, class-wise thresholds for distinguishing between ID and OOD samples are continuously revised through training, leading to more accurate separation of ID and OOD samples as training proceeds. Finally, when all of the losses are used together (UOTA), the separation of ID and OOD samples becomes more precise, and the performance of ID image classification greatly improves, resulting in the highest performance.

**Robustness on varying the ratio between ID and OOD samples.** We conduct openness experiments on the Office-31 dataset to observe whether UOTA shows improved performance over "Zero-shot" given different numbers of ID and OOD samples. To do that, we vary the number of known and unknown classes used as labels for ID and OOD samples, respectively. We use the average HOS scores of all domains in the dataset. As shown in Figure 5, regardless of the varying number of known or unknown classes, UOTA (green) consistently outperforms "Zero-shot" (red) by a significant margin.

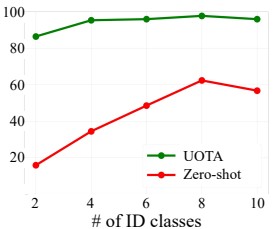 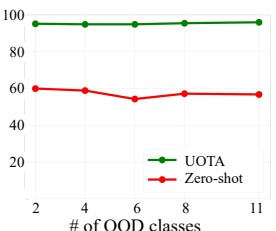

(a) Different # of ID classes.  (b) Different # of OOD classes.

Figure 5: **Robustness on different number of ID and OOD classes.** Using the Office-31, we evaluate the performance of "Zero-shot" (pre-trained CLIP) and UOTA according to the varying number of ID and OOD classes. In both settings, UOTA (green) consistently shows higher HOS scores than "Zero-shot" (red) regardless of the number of ID and OOD classes.

**Robustness on different backbones.** We also conduct ablation on backbones (*i.e.*, feature extractors) to observe if UOTA consistently improves "Zero-shot" when given backbones with different scales. We compare three different backbones, denoted as (1) "ViT-B/16", (2) "ViT-B/32", and (3) "ViT-L/14" (our default backbone). We use the Office-31, Office-Home, and VisDA datasets, with the HOS score as the evaluation metric. As presented in Table 5, UOTA consistently shows improved average HOS scores compared to "Zero-shot" and presents state-of-the-art performance, even when the backbone is changed.

Table 5: **Ablation on different backbones.** UOTA achieves higher HOS scores than "Zero-shot" for all datasets and target domains, regardless of the scale of its backbone. The bold results represent the best scores, while the underlined one is the second-best score.

| METHOD | OFFICE-HOME | | | | | OFFICE-31 | | | | VISDA |
|---|---|---|---|---|---|---|---|---|---|---|
| | P | R | C | A | AVG. | A | W | D | AVG. | R |
| ZERO-SHOT-VIT-B/16 | 61.6 | 66.2 | 67.7 | 69.8 | 66.3 | 53.7 | 49.4 | 55.2 | 52.8 | 85.2 |
| ZERO-SHOT-VIT-B/32 | 65.6 | 68.1 | 67.2 | 69.5 | 67.6 | 52.7 | 66.1 | 63.6 | 60.8 | 84.0 |
| ZERO-SHOT-VIT-L/14 | 57.4 | 63.9 | 63.1 | 69.2 | 63.4 | 48.0 | 57.0 | 65.3 | 56.8 | 83.1 |
| UOTA-VIT-B/16 | 87.0 | 87.4 | 79.1 | 79.1 | 83.2 | 89.6 | **96.0** | 98.1 | 94.6 | 89.7 |
| UOTA-VIT-B/32 | 84.2 | 86.2 | 75.1 | 75.9 | 80.4 | 89.5 | 87.2 | 88.6 | 88.4 | 85.3 |
| **UOTA-VIT-L/14 (OURS)** | **92.8** | **92.2** | **85.4** | **83.7** | **88.5** | **93.8** | 94.7 | **99.3** | **96.0** | **93.7** |

## 5  CONCLUSION

In conclusion, we address the challenge of building a reliable image classification model in real-world scenarios by leveraging large amounts of easily collectible, unlabeled data in the wild, including both task-relevant (ID) and task-irrelevant (OOD) data. To achieve this, we propose Unsupervised Open-Set Task Adaptation (UOTA), a simple yet effective algorithm that substantially improves the capability of pre-trained CLIP to perform OOD detection as well as ID classification under the *open-set unlabeled data setting*. Our work provides a promising direction for utilizing unlabeled data in real-world scenarios, potentially overcoming the cost and practical limitations of human-annotated data, and enhancing the transferability of zero-shot models.

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

# A APPENDIX

We provide supplementary materials for "Unsupervised Open-Set Task Adaptation Using a Vision-Language Foundation Model" in this document.

# B HYPERPARAMETERS

Tab. 1 presents the hyperparameters utilized for "Zero-shot (pre-trained CLIP)" and UOTA in our experiment. While some adjustments were made to a few hyperparameters for specific datasets, it is noteworthy that the experiment was mostly conducted without any significant hyperparameter tuning. In fact, slight differences in hyperparameters did not have a considerable impact on the experimental results. This demonstrates our model's robustness on hyperparameters.

Table 1: **List of hyperparameters.**

| Hyper-parameter | Office-31 | Office-Home | VisDA | DomainNet | Office-31 (Oracle) | Office-Home (Oracle) | VisDA (Oracle) | DomainNet (Oracle) |
|---|---|---|---|---|---|---|---|---|
| batch size | 32 | 32 | 32 | 32 | 32 | 32 | 32 | 32 |
| optimizer | AdamW | AdamW | AdamW | AdamW | AdamW | AdamW | AdamW | AdamW |
| learning rate | 1e-5 | 1e-5 | 1e-5 | 1e-5 | 1e-5 | 1e-5 | 1e-5 | 1e-5 |
| $\delta_{\text{in}}$ | 0.95 | 0.95 | 0.95 | 0.95 | 0.95 | 0.95 | 0.95 | 0.95 |
| $\delta_{\text{out}}$ | 0.5 | 0.5 | 0.8 | 0.8 | 0.5 | 0.5 | 0.8 | 0.8 |
| $\gamma$ | 4.0 | 4.0 | 4.0 | 4.0 | 4.0 | 4.0 | 4.0 | 4.0 |
| $\omega$ | 1.0 | 1.0 | 1.0 | 1.0 | 1.0 | 10.0 | 10.0 | 1.0 |
| $\delta_{\text{ood}}$ | 0.6 | 0.6 | 0.6 | 0.6 | 0.6 | 0.6 | 0.6 | 0.6 |

# C ADDITIONAL QUALITATIVE ANALYSIS

## C.1 HISTOGRAM VISUALIZATION

In this experiment, we use histograms to visualize how effectively UOTA distinguishes OOD samples from ID samples in comparison to "Zero-shot." The horizontal axis of the histogram represents the maximum probability scores, while the vertical axis denotes the number of samples. We use the Office-31, Office-Home, and VisDA datasets. With the histogram visualization, we can observe that UOTA computes lower maximum probability scores for unknown class (OOD) samples and higher maximum probability scores for known class (ID) samples, thereby exhibiting a significantly improved performance in differentiating OOD data from ID data when compared to "Zero-shot." Since the case of using ViT-L/14 as the backbone and Office-31 as a dataset has already been illustrated in Figure 3 of the main paper, we omit it here. Based on the results presented in Figure 1, 2, 3, 4, 5, and 6, we can confirm that, regardless of the dataset or backbone type, UOTA consistently segregates OODs from IDs more effectively than "Zero-shot."

## C.2 T-SNE VISUALIZATION

In this section, by utilizing t-SNE visualizations, we experimentally demonstrate that UOTA consistently exhibits significantly improved ID and OOD discrimination capabilities compared to the pre-trained CLIP "Zero-shot (pre-trained CLIP)", irrespective of the backbone employed. We use the Office-31 and Office-Home datasets, and each datapoint in the t-SNE figure represents the classification probability vector (described in Equation 1 of the main paper), as explained in Section 4.3 of the main paper. The backbones used are (1) ViT-B/16, (2) ViT-B/32, and (3) ViT-L/14 (our default backbone). Since the case of using ViT-L/14 as the backbone and Office-31 as a dataset has already been illustrated in Figure 3 of the main paper, we omit it here. Based on the results illustrated in Figure 7, 8, 9, 10, and 11, we can confirm that, regardless of the dataset or backbone type, UOTA consistently separates OODs from IDs more effectively than "Zero-shot" while simultaneously clustering OOD features more cohesively.

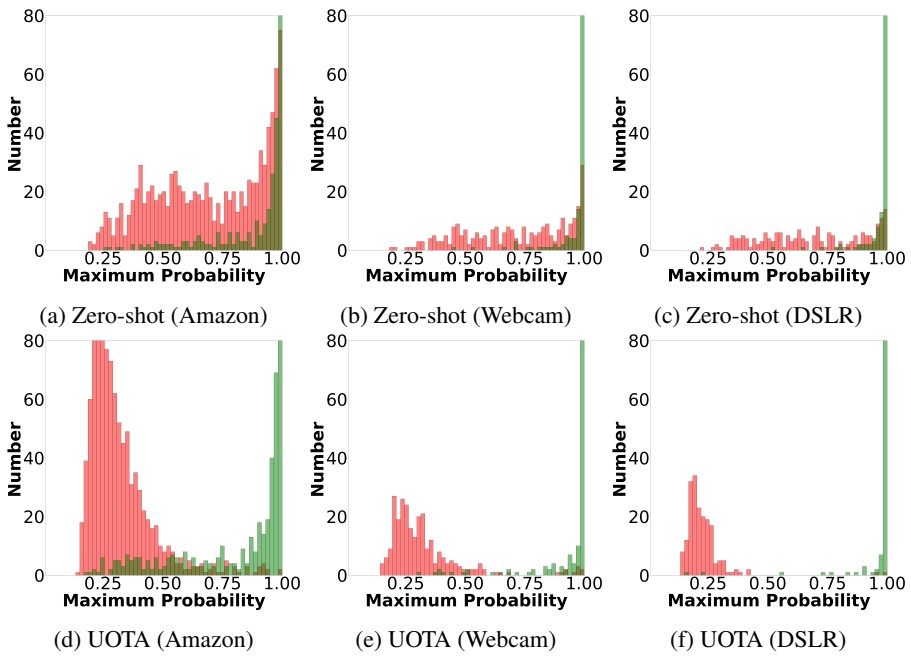

Figure 1: **Histogram visualization on Office-31 using ViT-B/16.** When employing the Office-31 dataset and using ViT-B/16 as the backbone for histogram visualization, the results of UOTA exhibit a more clear separation of OODs (red) from IDs (green) by producing lower maximum probability scores for OODs and higher maximum probability scores for IDs. We conducted experiments on three distinct domains, named (1) "Amazon", (2) "Webcam", and (3) "DSLR."

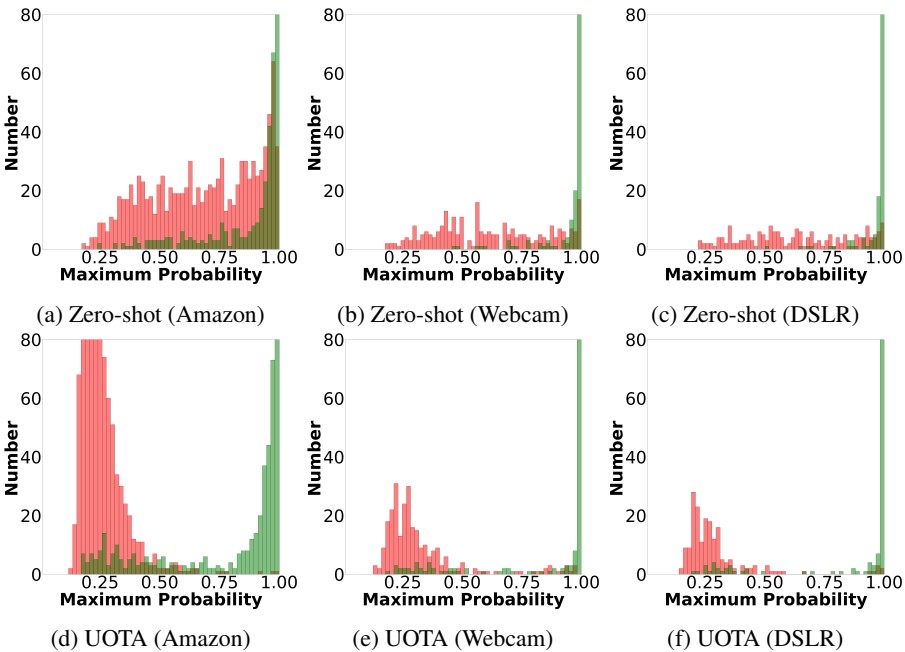

Figure 2: **Histogram visualization on Office-31 using ViT-B/32.** When employing the Office-31 dataset and using ViT-B/32 as the backbone for histogram visualization, the results of UOTA exhibit a more distinct separation of OODs (red) from IDs (green) by computing lower maximum probability scores for OODs and higher maximum probability scores for IDs. We conducted experiments on three distinct domains, named (1) "Amazon", (2)"Webcam", and (3) "DSLR."

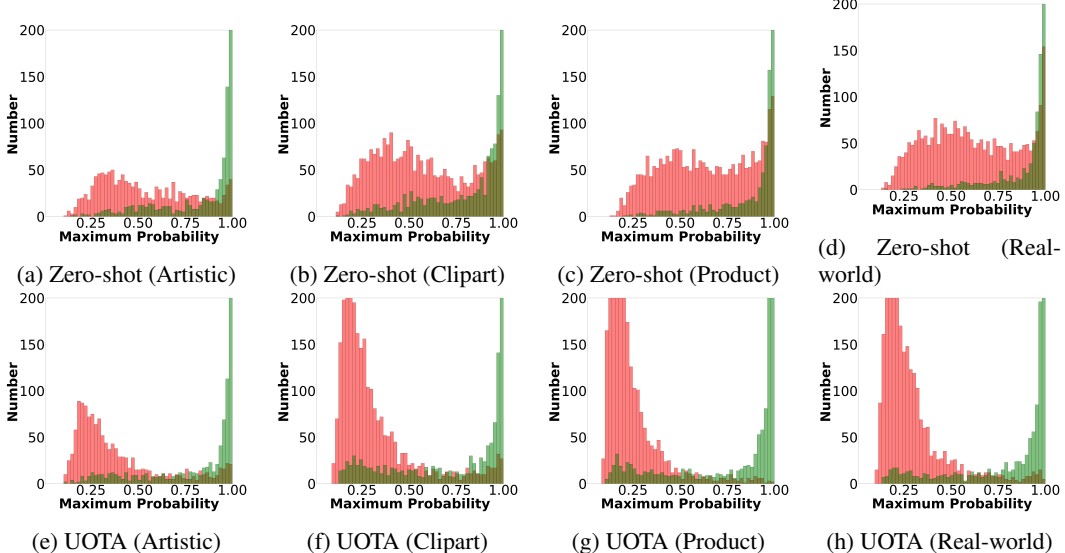

Figure 3: **Histogram visualization on Office-Home using ViT-B/16.** When employing the Office-Home dataset and using ViT-B/16 as the backbone for histogram visualization, the results of UOTA exhibit a more distinct separation of OOD samples (red) from ID samples (green) compared to the "Zero-shot" by computing lower maximum probability scores for OODs while producing lower maximum probability scores for IDs. We conducted experiments on four distinct domains, named (1) "Artistic", (2) "Clipart", (3) "Product", and (4) "Real-world."

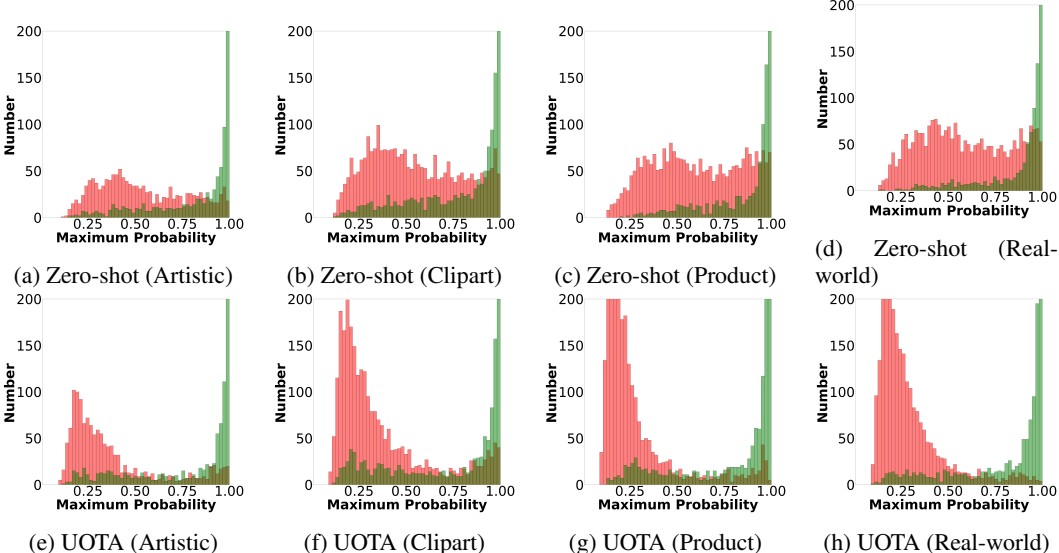

Figure 4: **Histogram visualization on Office-Home using ViT-B/32.** When employing the Office-Home dataset and using ViT-B/32 as the backbone for histogram visualization, the results of UOTA exhibit a more noticeable discrimination of OOD samples (red) from ID samples (green) compared to the "Zero-shot." We conducted experiments on four distinct domains, named (1) "Artistic", (2) "Clipart", (3) "Product", and (4) "Real-world."

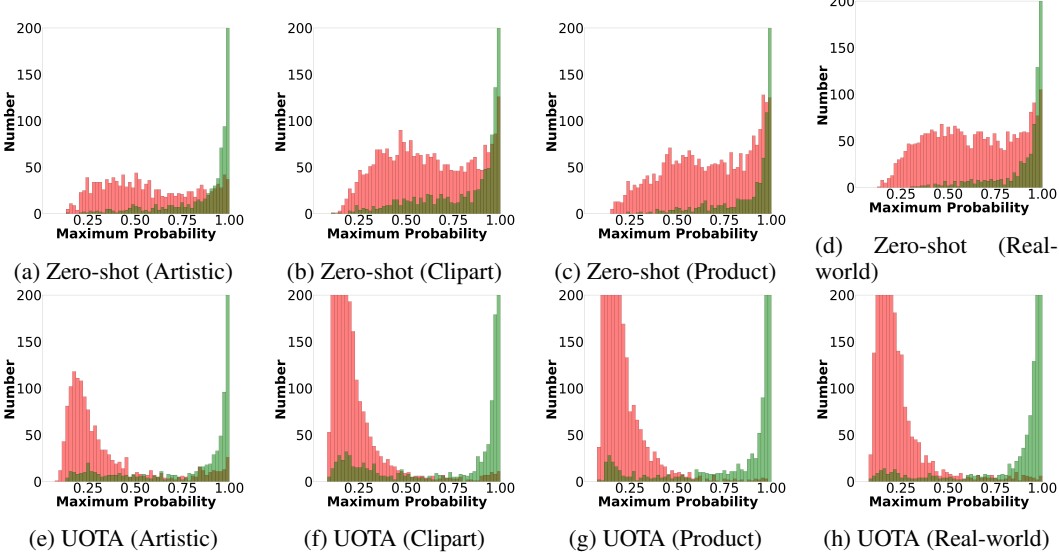

Figure 5: **Histogram visualization on Office-Home using ViT-L/14.** When employing the Office-Home dataset and using ViT-L/14 (our default backbone) as the backbone for histogram visualization, the results of UOTA exhibit a more clear segregation of OOD samples (red) from ID samples (green) compared to the "Zero-shot." We conducted experiments on four distinct domains, named (1) "Artistic", (2) "Clipart", (3) "Product", and (4) "Real-world."

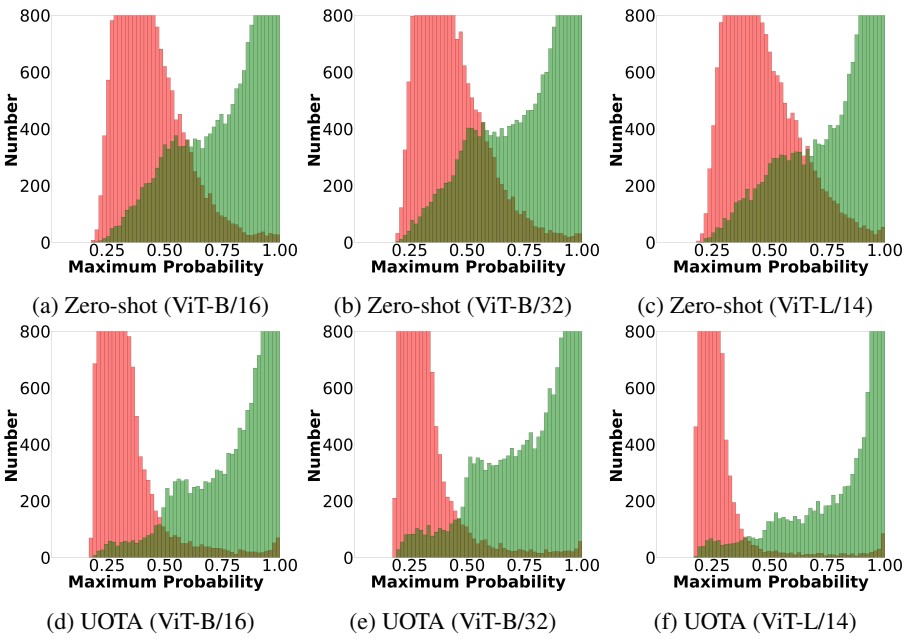

Figure 6: **Histogram visualization on VisDA using ViT-B/16, ViT-B/32, and ViT-L/14.** When employing the VisDA dataset and using ViT-B/16, ViT-B/32, and ViT-L/14 as the backbones for histogram visualization, the results of UOTA exhibit a more clear separation of OODs (red) from IDs (green), regardless of the type of backbone employed. (a), (d) are results of ViT-B/16; (b), (e) are results of ViT-B/32; and (c), (f) are results of ViT-L/14.

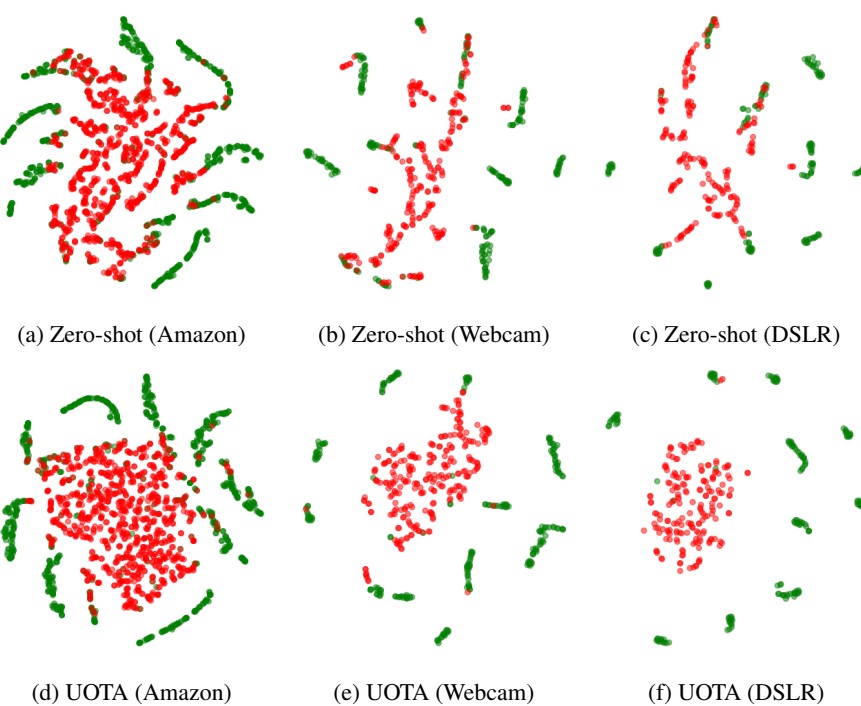

(a) Zero-shot (Amazon)          (b) Zero-shot (Webcam)          (c) Zero-shot (DSLR)

(d) UOTA (Amazon)          (e) UOTA (Webcam)          (f) UOTA (DSLR)

Figure 7: **t-SNE visualization on Office-31 using ViT-B/16.** When employing the Office-31 dataset and using ViT-B/16 as the backbone for t-SNE visualization, the results of UOTA exhibit a more clear separation of OODs (red) from IDs (green) and more tight clustering of OODs compared to the "Zero-shot." We conducted experiments on three distinct domains, named (1) "Amazon", (2) "Webcam", and (3) "DSLR."

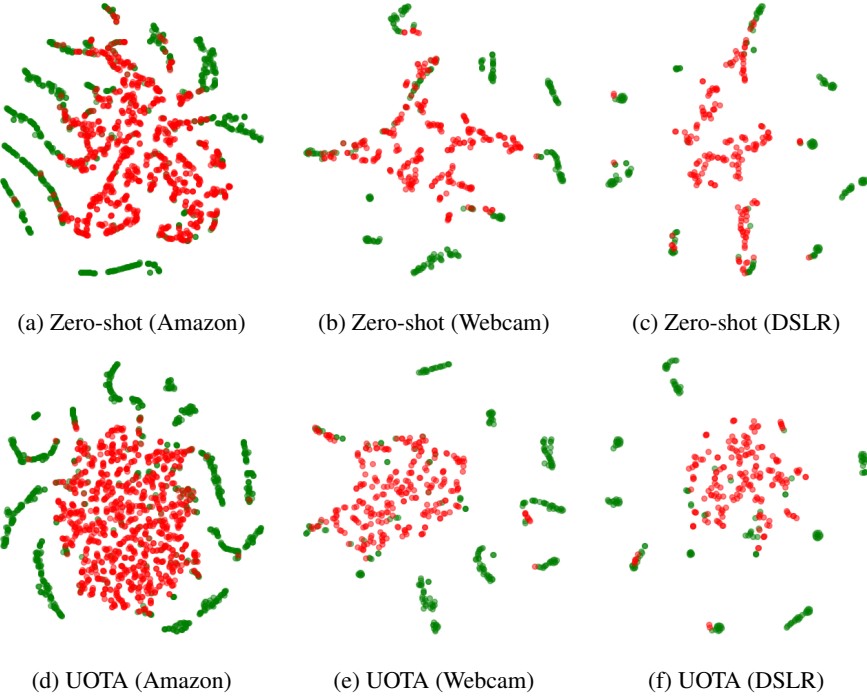

(a) Zero-shot (Amazon)          (b) Zero-shot (Webcam)          (c) Zero-shot (DSLR)

(d) UOTA (Amazon)          (e) UOTA (Webcam)          (f) UOTA (DSLR)

Figure 8: **t-SNE visualization on Office-31 using ViT-B/32.** When employing the Office-31 dataset and using ViT-B/32 as the backbone for t-SNE visualization, the results of UOTA exhibit a more distinct separation of OOD samples (red) from ID data (green) and more tight clustering of OODs compared to the "Zero-shot." We conducted experiments on three distinct domains, named (1) "Amazon", (2) "Webcam", and (3) "DSLR."

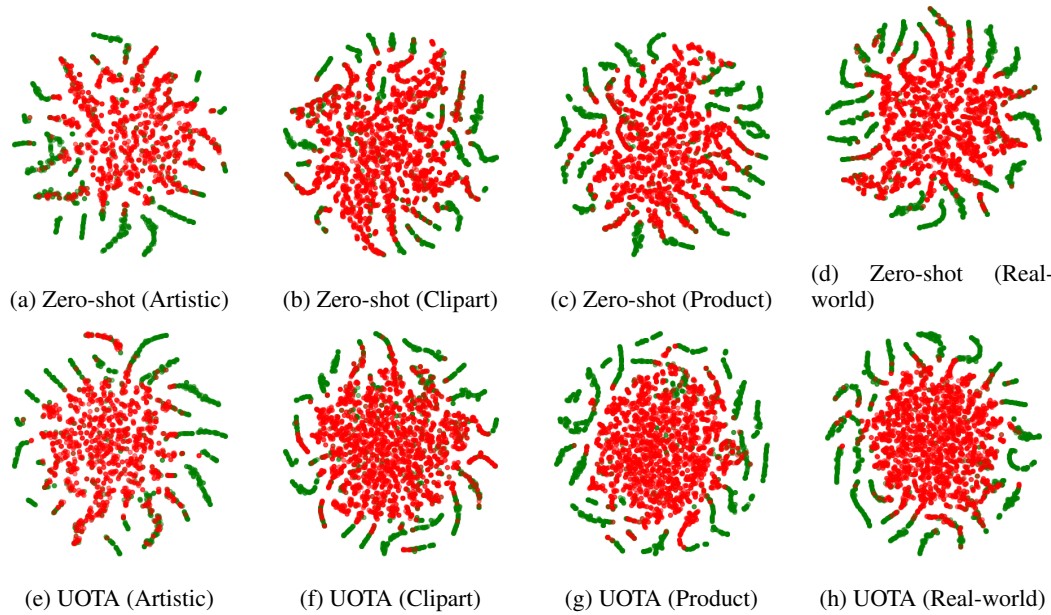

(a) Zero-shot (Artistic)   (b) Zero-shot (Clipart)   (c) Zero-shot (Product)   (d) Zero-shot (Real-world)

(e) UOTA (Artistic)   (f) UOTA (Clipart)   (g) UOTA (Product)   (h) UOTA (Real-world)

Figure 9: **t-SNE visualization on Office-Home using ViT-B/16.** When employing the Office-Home dataset and using ViT-B/16 as the backbone for t-SNE visualization, the results of UOTA exhibit a more distinct separation of OOD samples (red) from ID samples (green) and more tight clustering of OODs compared to the "Zero-shot." We conducted experiments on four distinct domains, named (1) "Artistic", (2) "Clipart", (3) "Product", and (4) "Real-world."

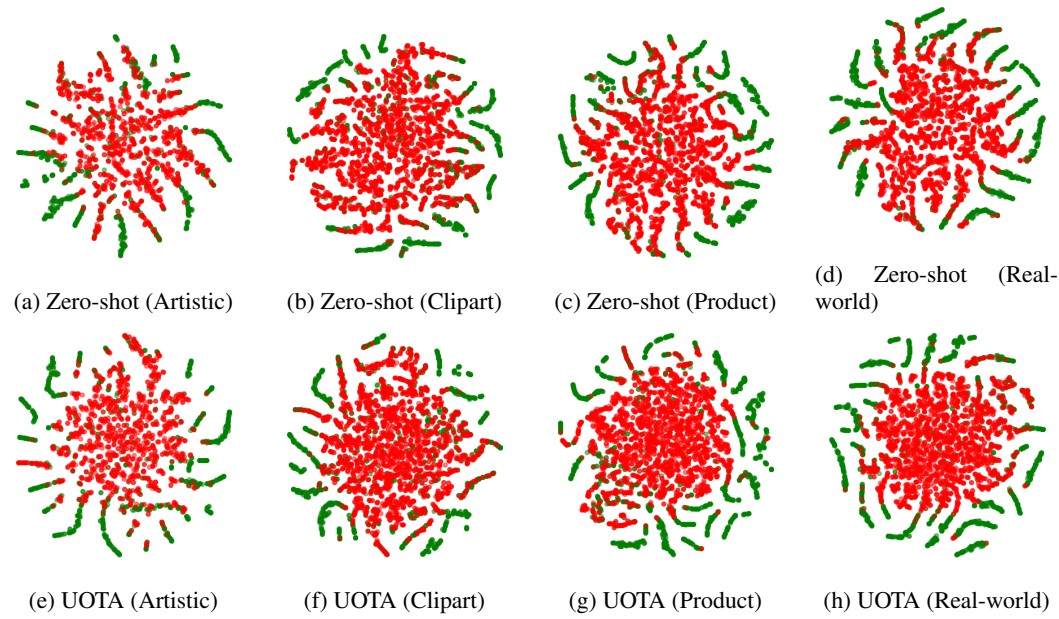

(a) Zero-shot (Artistic)   (b) Zero-shot (Clipart)   (c) Zero-shot (Product)   (d) Zero-shot (Real-world)

(e) UOTA (Artistic)   (f) UOTA (Clipart)   (g) UOTA (Product)   (h) UOTA (Real-world)

Figure 10: **t-SNE visualization on Office-Home using ViT-B/32.** When employing the Office-Home dataset and using ViT-B/32 as the backbone for t-SNE visualization, the results of UOTA exhibit a more distinct separation of OODs (red) from IDs (green) and more tight clustering of OODs compared to the "Zero-shot." We conducted experiments on four distinct domains, named (1) "Artistic", (2) "Clipart", (3) "Product", and (4) "Real-world."

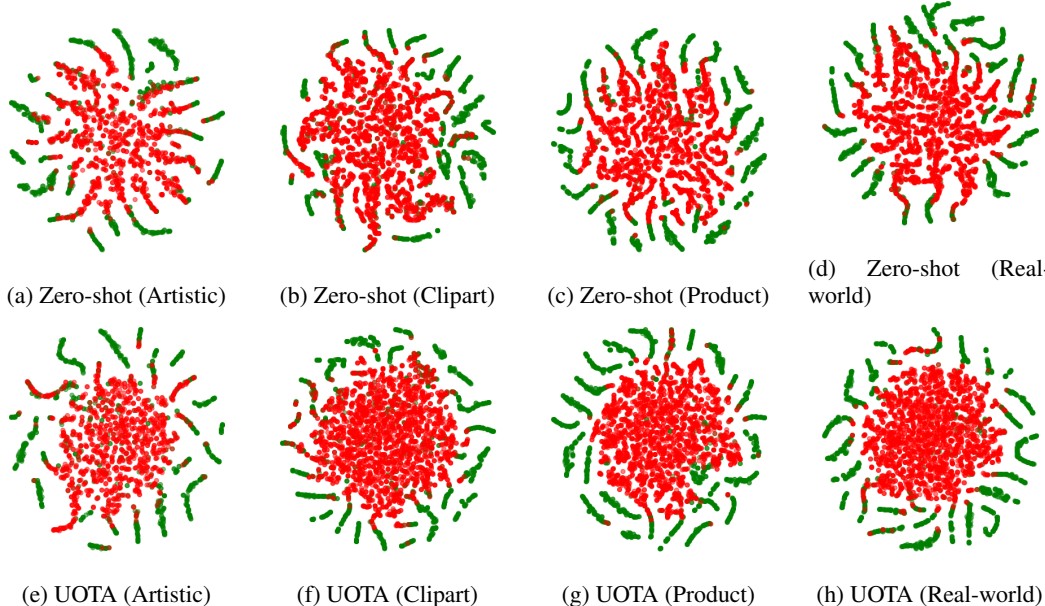

(a) Zero-shot (Artistic)  (b) Zero-shot (Clipart)  (c) Zero-shot (Product)  (d) Zero-shot (Real-world)

(e) UOTA (Artistic)  (f) UOTA (Clipart)  (g) UOTA (Product)  (h) UOTA (Real-world)

Figure 11: **t-SNE visualization on Office-Home using ViT-L/14.** When employing the Office-Home dataset and using ViT-L/14 as the backbone for t-SNE visualization, the results of UOTA exhibit a more distinct separation of OOD data (red) from ID data (green) and more tight clustering of OODs compared to the "Zero-shot." We conducted experiments on four distinct domains, named (1) "Artistic", (2) "Clipart", (3) "Product", and (4) "Real-world."

# D   SUPPLEMENTAL EXPERIMENT RESULTS ON OFFICE-31, OFFICE-HOME, AND VISDA

Table 2: Additional results on Office-31.

| METHOD | W | | | D | | | A | | | D | | | A | | | W | | | AVG | | |
| | A | | | A | | | W | | | W | | | D | | | D | | | | | |
| | OS* | UNK | HOS | OS* | UNK | HOS | OS* | UNK | HOS | OS* | UNK | HOS | OS* | UNK | HOS | OS* | UNK | HOS | OS* | UNK | HOS |
|---|---|---|---|---|---|---|---|---|---|---|---|---|---|---|---|---|---|---|---|---|---|
| DANN | 72.1 | 73.1 | 72.6 | 72.9 | 74.5 | 73.7 | 87.4 | 55.7 | 68.1 | 99.3 | 77.0 | 86.7 | 90.8 | 59.2 | 71.5 | 100.0 | 70.2 | 82.5 | 87.1 | 68.3 | 75.9 |
| CDAN | 72.8 | 69.3 | 71.0 | 74.9 | 70.6 | 72.7 | 90.3 | 50.7 | 64.9 | 99.6 | 73.2 | 84.3 | 92.2 | 52.4 | 66.8 | 100.0 | 67.3 | 80.5 | 88.3 | 63.9 | 73.4 |
| OSBP | 73.0 | 74.4 | 73.7 | 76.1 | 72.3 | 75.1 | 86.8 | 79.2 | 82.7 | 97.7 | 96.7 | 97.2 | 90.5 | 75.5 | 82.4 | 99.1 | 84.2 | 91.1 | 87.2 | 80.4 | 83.7 |
| STA | 66.2 | 68.0 | 66.1 | 83.1 | 65.9 | 73.2 | 86.7 | 67.6 | 75.9 | 94.1 | 55.5 | 69.8 | 91.0 | 63.9 | 75.0 | 84.9 | 67.8 | 75.2 | 84.3 | 64.8 | 72.5 |
| PGL | 80.8 | 61.8 | 70.1 | 80.6 | 61.2 | 69.5 | 82.7 | 67.9 | 74.6 | 87.5 | 68.1 | 76.5 | 82.1 | 65.4 | 72.8 | 82.8 | 64.0 | 72.2 | 82.7 | 64.7 | 72.6 |
| ROS | 69.7 | 86.6 | 77.2 | 74.8 | 81.2 | 77.9 | 88.4 | 76.7 | 82.1 | 99.3 | 93.0 | 96.0 | 87.5 | 77.8 | 82.4 | 100.0 | 99.4 | 99.7 | 86.6 | 85.8 | 85.9 |
| DANCE | 83.7 | 60.6 | 70.2 | 85.3 | 53.6 | 65.8 | 98.7 | 50.7 | 66.9 | 100.0 | 66.8 | 80.0 | 96.5 | 55.9 | 70.7 | 100.0 | 73.7 | 84.8 | 94.0 | 60.2 | 73.1 |
| DCC | - | - | 84.4 | - | - | 85.5 | - | - | 87.1 | - | - | 91.2 | - | - | 85.5 | - | - | 87.1 | - | - | 86.8 |
| OSLPP | 78.9 | 78.5 | 78.7 | 82.1 | 76.6 | 79.3 | 89.5 | 88.4 | 89.0 | 96.9 | 88.0 | 92.3 | 92.6 | 90.4 | 91.5 | 95.8 | 91.5 | 93.6 | 89.3 | 85.6 | 87.4 |
| UADAL | 67.4 | 88.4 | 76.5 | 73.3 | 87.3 | 79.7 | 84.3 | 94.5 | 89.1 | 99.3 | 96.3 | 97.8 | 85.1 | 87.0 | 86.0 | 99.5 | 99.4 | 99.5 | 84.8 | 92.1 | 88.1 |
| cUADAL | 65.6 | 87.8 | 75.1 | 74.2 | 87.8 | 80.5 | 85.5 | 95.1 | 90.1 | 98.7 | 97.7 | 98.2 | 85.6 | 90.4 | 87.9 | 99.3 | 99.4 | 99.4 | 84.8 | 93.0 | 88.5 |
| SHOT | 72.2 | 80.1 | 75.9 | 75.5 | 72.5 | 74.0 | 74.5 | 64.4 | 69.1 | 96.7 | 79.4 | 87.2 | 82.0 | 56.9 | 67.2 | 98.8 | 87.2 | 92.7 | 83.3 | 73.4 | 77.7 |
| AaD | 70.8 | 78.2 | 73.9 | 69.8 | 77.4 | 73.0 | 74.6 | 83.5 | 78.3 | 90.2 | 92.5 | 91.2 | 75.3 | 80.9 | 77.7 | 92.1 | 95.2 | 93.5 | 78.8 | 84.6 | 81.3 |
| UOTA | 89.6 | 98.5 | 93.8 | 89.6 | 98.5 | 93.8 | 90.0 | 100.0 | 94.7 | 90.0 | 100.0 | 94.7 | 98.7 | 100.0 | 99.3 | 98.7 | 100.0 | 99.3 | 92.8 | 99.5 | 96.0 |

Table 3: Additional results on Office-Home.

| METHOD | R | | | C | | | A | | | P | | | C | | | A | | |
| | | | | | | | | | | | | | | R | | | | | |
| | | | | P | | | | | | | | | | | | | | | |
| | OS* | UNK | HOS | OS* | UNK | HOS | OS* | UNK | HOS | OS* | UNK | HOS | OS* | UNK | HOS | OS* | UNK | HOS |
|---|---|---|---|---|---|---|---|---|---|---|---|---|---|---|---|---|---|---|
| CDAN | 70.9 | 64.6 | 67.6 | 51.6 | 76.8 | 61.7 | 61.7 | 68.8 | 65.1 | 69.8 | 69.7 | 69.7 | 61.5 | 73.7 | 67.1 | 75.2 | 66.7 | 70.7 |
| OSBP | 76.3 | 68.6 | 72.3 | 67.0 | 62.7 | 64.7 | 71.8 | 59.8 | 65.2 | 76.2 | 71.7 | 73.9 | 72.0 | 69.2 | 70.6 | 79.3 | 67.5 | 72.9 |
| STA | 77.1 | 55.4 | 64.5 | 61.8 | 59.1 | 60.4 | 68.0 | 48.4 | 54.0 | 67.0 | 66.7 | 66.8 | 67.0 | 66.7 | 66.8 | 78.6 | 60.4 | 68.3 |
| PGL | 84.8 | 38.0 | 52.5 | 73.9 | 24.5 | 36.8 | 78.9 | 32.1 | 45.6 | 84.8 | 27.6 | 41.6 | 70.2 | 33.8 | 45.6 | 87.7 | 40.9 | 55.8 |
| ROS | 72.0 | 80.0 | 75.7 | 59.8 | 71.6 | 65.2 | 68.4 | 70.3 | 69.3 | 70.8 | 78.4 | 74.4 | 65.3 | 72.2 | 68.6 | 75.8 | 77.2 | 76.5 |
| DANCE | 86.2 | 29.6 | 44.0 | 76.3 | 32.8 | 45.9 | 84.0 | 35.4 | 49.8 | 86.5 | 27.1 | 41.2 | 83.9 | 18.4 | 30.2 | 89.8 | 25.3 | 39.4 |
| DCC | - | - | 62.7 | - | - | 66.6 | - | - | 67.4 | - | - | 64.0 | - | - | 67.0 | - | - | 80.6 |
| LGU | 83.2 | 46.8 | 59.9 | 71.7 | 4.1 | 7.8 | 80.5 | 49.3 | 61.2 | 41.2 | 55.0 | | 77.6 | 46.4 | 58.1 | 86.5 | 47.5 | 61.3 |
| OSLPP | 78.4 | 70.8 | 74.4 | 61.6 | 73.3 | 66.9 | 72.5 | 73.1 | 72.8 | 77.0 | 71.2 | 74.0 | 67.2 | 73.9 | 70.4 | 80.1 | 69.4 | 74.3 |
| UADAL | 77.4 | 76.2 | 76.8 | 62.1 | 78.8 | 69.5 | 69.1 | 72.5 | 70.8 | 71.6 | 83.1 | 76.9 | 69.1 | 78.3 | 73.4 | 81.3 | 73.7 | 77.4 |
| cUADAL | 77.8 | 75.6 | 76.7 | 61.1 | 77.4 | 68.3 | 69.4 | 73.9 | 71.6 | 71.2 | 83.4 | 76.8 | 69.3 | 76.3 | 72.6 | 82.2 | 73.3 | 77.5 |
| SHOT | 84.4 | 28.2 | 42.3 | 77.5 | 27.2 | 40.2 | 81.8 | 26.3 | 39.8 | 85.8 | 31.6 | 46.2 | 80.0 | 25.9 | 39.1 | 87.5 | 32.1 | 47.0 |
| AaD | 69.7 | 70.6 | 70.1 | 59.5 | 63.5 | 61.4 | 64.6 | 69.4 | 66.9 | 68.4 | 72.8 | 70.6 | 67.4 | 68.3 | 67.8 | 73.1 | 66.9 | 69.9 |
| UOTA | 88.2 | 97.9 | 92.8 | 88.2 | 97.9 | 92.8 | 88.2 | 97.9 | 92.8 | 88.6 | 96.1 | 92.2 | 88.6 | 96.1 | 92.2 | 88.6 | 96.1 | 92.2 |

| METHOD | P | | | R | | | A | | | P | | | R | | | C | | | AVG | | |
| | | | | | | | | | | | | | | A | | | | | | | |
| | | | | C | | | | | | | | | | | | | | | | | |
| | OS* | UNK | HOS | OS* | UNK | HOS | OS* | UNK | HOS | OS* | UNK | HOS | OS* | UNK | HOS | OS* | UNK | HOS | OS* | UNK | HOS |
|---|---|---|---|---|---|---|---|---|---|---|---|---|---|---|---|---|---|---|---|---|---|
| DANN | 30.1 | 86.3 | 44.6 | 37.1 | 80.9 | 50.9 | 37.1 | 82.7 | 51.2 | 42.4 | 83.9 | 56.3 | 56.8 | 77.1 | 65.4 | 43.8 | 84.3 | 57.6 | 52.6 | 77.1 | 60.7 |
| CDAN | 33.1 | 82.4 | 47.2 | 40.3 | 75.8 | 52.7 | 39.7 | 78.9 | 52.9 | 45.8 | 81.2 | 58.6 | 59.8 | 73.6 | 66.0 | 44.9 | 82.8 | 58.2 | 54.5 | 74.6 | 61.4 |
| OSBP | 44.5 | 66.3 | 53.2 | 48.0 | 63.0 | 54.5 | 50.2 | 61.1 | 55.1 | 59.1 | 66.1 | 63.2 | 66.1 | 67.3 | 66.7 | 59.4 | 70.3 | 64.3 | 64.1 | 66.3 | 64.7 |
| STA | 44.2 | 67.1 | 53.2 | 49.9 | 61.1 | 54.5 | 46.0 | 72.3 | 55.8 | 54.2 | 72.4 | 61.9 | 67.5 | 66.7 | 67.1 | 51.4 | 65.0 | 57.4 | 61.8 | 63.3 | 61.1 |
| PGL | 59.2 | 38.4 | 46.6 | 68.8 | 0.0 | 0.0 | 63.3 | 19.1 | 29.3 | 73.7 | 34.7 | 47.2 | 81.5 | 6.1 | 11.4 | 85.9 | 5.3 | 10.0 | 76.1 | 25.0 | 35.2 |
| ROS | 46.5 | 71.2 | 56.3 | 51.5 | 73.0 | 60.4 | 50.6 | 74.1 | 60.1 | 57.3 | 64.3 | 60.6 | 67.0 | 70.8 | 68.8 | 53.6 | 65.5 | 58.9 | 61.6 | 72.4 | 66.2 |
| DANCE | 48.2 | 67.4 | 55.7 | 60.1 | 41.3 | 48.3 | 54.4 | 53.7 | 53.1 | 70.7 | 43.9 | 54.2 | 79.2 | 16.7 | 27.5 | 72.9 | 28.4 | 40.9 | 74.4 | 35.0 | 44.2 |
| DCC | - | - | 52.8 | - | - | 76.9 | - | - | 52.9 | - | - | 59.5 | - | - | 56.0 | - | - | 49.8 | - | - | 64.2 |
| LGU | 54.5 | 18.1 | 27.2 | 63.4 | 29.6 | 40.4 | 58.6 | 32.6 | 41.9 | 69.1 | 50.9 | 58.6 | 77.5 | 48.9 | 60.0 | 67.2 | 30.8 | 42.2 | 72.7 | 38.9 | 50.7 |
| OSLPP | 53.1 | 67.1 | 59.3 | 54.4 | 64.3 | 59.0 | 55.9 | 67.1 | 61.0 | 54.6 | 76.2 | 63.6 | 60.8 | 75.0 | 67.2 | 49.6 | 79.0 | 60.9 | 63.8 | 71.7 | 67.0 |
| UADAL | 43.4 | 81.5 | 56.6 | 51.1 | 74.5 | 60.6 | 54.9 | 74.7 | 63.2 | 50.5 | 83.7 | 63.0 | 66.7 | 78.6 | 72.1 | 53.5 | 80.5 | 64.2 | 62.6 | 78.0 | 68.7 |
| cUADAL | 41.2 | 80.7 | 54.6 | 51.8 | 71.1 | 59.9 | 55.0 | 75.6 | 63.6 | 50.9 | 82.4 | 62.9 | 66.8 | 79.6 | 72.6 | 53.8 | 82.0 | 65.0 | 62.5 | 77.6 | 68.5 |
| SHOT | 59.3 | 31.0 | 40.8 | 65.3 | 28.9 | 40.1 | 67.0 | 28.0 | 39.5 | 66.3 | 51.1 | 57.7 | 73.5 | 50.6 | 59.9 | 66.8 | 46.2 | 54.6 | 74.6 | 33.9 | 45.6 |
| AaD | 45.4 | 72.8 | 55.9 | 49.0 | 69.6 | 57.5 | 50.7 | 66.4 | 57.6 | 47.3 | 82.4 | 60.1 | 54.5 | 79.0 | 64.6 | 48.2 | 81.1 | 60.5 | 58.2 | 71.9 | 63.6 |
| UOTA | 76.9 | 95.9 | 85.4 | 76.9 | 95.9 | 85.4 | 76.9 | 95.9 | 85.4 | 79.1 | 88.9 | 83.7 | 79.1 | 88.9 | 83.7 | 79.1 | 88.9 | 83.7 | 83.2 | 94.7 | 88.5 |

Table 4: Additional results on VisDA.

| METHOD | VISDA | | |
| | OS* | UNK | HOS |
|---|---|---|---|
| STA | 63.9 | 84.2 | 72.7 |
| OSBP | 59.2 | 85.1 | 69.8 |
| PGL | 82.8 | 68.1 | 74.7 |
| DCC | 68.0 | 73.6 | 70.7 |
| UADAL | 61.1 | 93.3 | 75.3 |
| cUADAL | 64.3 | 92.6 | 75.9 |
| SHOT | 44.6 | 40.7 | 42.6 |
| AaD | 13.8 | 23.3 | 16.0 |
| UOTA | 89.4 | 98.4 | 93.7 |

In Tab. 2, 3, and 4, we provide additional results measuring the performance of UOTA and other existing models using OS* (accuracy over known classes) and UNK (accuracy of unknown classes). Note that, different from the HOS score, the OS* and UNK are biased evaluation metrics that do not simultaneously consider a model's ID classification and OOD detection capabilities. We employ Office-31, Office-Home, and VisDA as datasets. We utilize OSDA models (DANN, CDAN, OSBP, STA, PGL, ROS, DANCE, DCC, OSLPP, UADAL, and cUADAL) and SF-OSDA models (SHOT and AaD) as comparison models. We conduct experiments using the best settings for each of these models on their respective datasets (e.g., use ResNet50 as a backbone for Office-31 and Office-Home, and use VGGNet as a backbone for VisDA). As mentioned in the main paper, UOTA

assumes a more constrained experimental environment that neither uses source models nor source data; however, it records state-of-the-art results for all datasets and target domains.

# E   SUPPLEMENTAL EXPERIMENT RESULTS ON DOMAINNET

We also provide the OS* and UNK score results for DomainNet. For comparison, we used SF-OSDA models (SHOT, AaD) and conducted experiments utilizing their best settings (e.g. using ResNet50 as a backbone). Tab. 5, 6, 7, 8, 9, and 10 display the results of experiments conducted with varying target domains.

Table 5: Additional results on DomainNet (Target domain: Quick draw).

| METHOD | C | | | I | | | P | | | R | | | S | | | AVG. | | |
|---|---|---|---|---|---|---|---|---|---|---|---|---|---|---|---|---|---|---|
| | OS* | UNK | HOS | OS* | UNK | HOS | OS* | UNK | HOS | OS* | UNK | HOS | OS* | UNK | HOS | OS* | UNK | HOS |
| SHOT | 11.1 | 38.6 | 17.2 | 3.3 | 21.1 | 5.7 | 8.7 | 34.9 | 13.9 | 9.8 | 39.8 | 15.7 | 3.9 | 34.1 | 6.9 | 7.4 | 33.7 | 12.3 |
| AaD | 13.3 | 62.7 | 21.3 | 4.6 | 60.0 | 7.5 | 6.8 | 63.0 | 11.3 | 9.6 | 60.4 | 15.7 | 12.6 | 67.2 | 20.4 | 9.4 | 62.7 | 15.3 |
| UOTA | 22.7 | 85.4 | 35.9 | 22.7 | 85.4 | 35.9 | 22.7 | 85.4 | 35.9 | 22.7 | 85.4 | 35.9 | 22.7 | 85.4 | 35.9 | 22.7 | 85.4 | 35.9 |

Table 6: Additional results on DomainNet (Target domain: Clipart).

| METHOD | Q | | | I | | | P | | | R | | | S | | | AVG. | | |
|---|---|---|---|---|---|---|---|---|---|---|---|---|---|---|---|---|---|---|
| | OS* | UNK | HOS | OS* | UNK | HOS | OS* | UNK | HOS | OS* | UNK | HOS | OS* | UNK | HOS | OS* | UNK | HOS |
| SHOT | 18.3 | 41.7 | 25.4 | 36.4 | 44.6 | 40.1 | 50.1 | 51.4 | 50.7 | 52.5 | 46.4 | 49.2 | 48.1 | 51.8 | 49.9 | 41.1 | 47.2 | 43.1 |
| AaD | 27.6 | 69.0 | 39.0 | 31.3 | 75.0 | 43.7 | 41.9 | 78.0 | 54.2 | 43.1 | 76.2 | 54.8 | 43.7 | 77.2 | 55.6 | 37.5 | 75.1 | 49.4 |
| UOTA | 89.4 | 76.5 | 82.4 | 89.4 | 76.5 | 82.4 | 89.4 | 76.5 | 82.4 | 89.4 | 76.5 | 82.4 | 89.4 | 76.5 | 82.4 | 89.4 | 76.5 | 82.4 |

Table 7: Additional results on DomainNet (Target domain: Infograph).

| METHOD | Q | | | C | | | P | | | R | | | S | | | AVG. | | |
|---|---|---|---|---|---|---|---|---|---|---|---|---|---|---|---|---|---|---|
| | OS* | UNK | HOS | OS* | UNK | HOS | OS* | UNK | HOS | OS* | UNK | HOS | OS* | UNK | HOS | OS* | UNK | HOS |
| SHOT | 2.5 | 38.8 | 4.6 | 20.0 | 49.6 | 28.5 | 21.1 | 48.5 | 29.4 | 21.1 | 43.1 | 28.4 | 18.2 | 45.5 | 26.0 | 16.6 | 45.1 | 23.4 |
| AaD | 3.9 | 75.7 | 6.1 | 17.3 | 16.6 | 27.8 | 19.2 | 83.6 | 30.4 | 20.1 | 80.1 | 31.3 | 17.1 | 82.0 | 27.4 | 15.5 | 67.6 | 24.6 |
| UOTA | 61.6 | 77.8 | 68.7 | 61.6 | 77.8 | 68.7 | 61.6 | 77.8 | 68.7 | 61.6 | 77.8 | 68.7 | 61.6 | 77.8 | 68.7 | 61.6 | 77.8 | 68.7 |

Table 8: Additional results on DomainNet (Target domain: Painting).

| METHOD | Q | | | C | | | I | | | R | | | S | | | AVG. | | |
|---|---|---|---|---|---|---|---|---|---|---|---|---|---|---|---|---|---|---|
| | OS* | UNK | HOS | OS* | UNK | HOS | OS* | UNK | HOS | OS* | UNK | HOS | OS* | UNK | HOS | OS* | UNK | HOS |
| SHOT | 5.5 | 24.6 | 9.0 | 50.9 | 32.5 | 39.7 | 36.0 | 42.8 | 39.1 | 48.8 | 45.9 | 47.3 | 41.4 | 30.1 | 34.9 | 36.5 | 35.2 | 34.0 |
| AaD | 14.4 | 70.5 | 23.1 | 39.9 | 76.8 | 52.2 | 31.4 | 70.0 | 43.0 | 44.0 | 74.2 | 55.0 | 39.5 | 75.8 | 51.7 | 33.8 | 73.5 | 30.9 |
| UOTA | 74.7 | 79.4 | 77.0 | 74.7 | 79.4 | 77.0 | 74.7 | 79.4 | 77.0 | 74.7 | 79.4 | 77.0 | 74.7 | 79.4 | 77.0 | 74.7 | 79.4 | 77.0 |

Table 9: Additional results on DomainNet (Target domain: Real).

| METHOD | Q | | | C | | | I | | | P | | | S | | | AVG. | | |
|---|---|---|---|---|---|---|---|---|---|---|---|---|---|---|---|---|---|---|
| | OS* | UNK | HOS | OS* | UNK | HOS | OS* | UNK | HOS | OS* | UNK | HOS | OS* | UNK | HOS | OS* | UNK | HOS |
| SHOT | 17.1 | 20.6 | 18.7 | 64.8 | 46.3 | 54.0 | 63.2 | 26.5 | 37.4 | 71.4 | 28.9 | 41.1 | 61.3 | 39.0 | 47.7 | 55.6 | 32.3 | 39.8 |
| AaD | 25.9 | 59.0 | 35.7 | 56.1 | 57.4 | 56.7 | 50.4 | 56.1 | 53.1 | 57.0 | 56.2 | 56.6 | 54.8 | 58.8 | 56.7 | 48.8 | 57.5 | 51.8 |
| UOTA | 89.0 | 81.8 | 85.2 | 89.0 | 81.8 | 85.2 | 89.0 | 81.8 | 85.2 | 89.0 | 81.8 | 85.2 | 89.0 | 81.8 | 85.2 | 89.0 | 81.8 | 85.2 |

Table 10: Additional results on DomainNet (Target domain: Sketch).

| METHOD | Q | | | C | | | I | | | P | | | R | | | AVG. | | |
|---|---|---|---|---|---|---|---|---|---|---|---|---|---|---|---|---|---|---|
| | OS* | UNK | HOS | OS* | UNK | HOS | OS* | UNK | HOS | OS* | UNK | HOS | OS* | UNK | HOS | OS* | UNK | HOS |
| SHOT | 16.1 | 41.6 | 23.2 | 43.5 | 47.2 | 45.3 | 30.9 | 44.9 | 36.6 | 42.9 | 44.4 | 43.6 | 33.7 | 38.6 | 36.0 | 33.4 | 43.3 | 36.9 |
| AaD | 21.3 | 71.0 | 32.2 | 37.6 | 76.7 | 50.1 | 24.3 | 70.5 | 35.7 | 36.0 | 78.8 | 49.0 | 34.6 | 71.4 | 46.3 | 30.8 | 73.7 | 42.6 |
| UOTA | 77.0 | 77.6 | 77.3 | 77.0 | 77.6 | 77.3 | 77.0 | 77.6 | 77.3 | 77.0 | 77.6 | 77.3 | 77.0 | 77.6 | 77.3 | 77.0 | 77.6 | 77.3 |

# F   PYTORCH-STYLE PSEUDOCODE FOR UOTA

---

**Algorithm 1** UOTA: PyTorch Pseudocode

---

```python
# img_1, img_2, img_encoder, txt_feat: View 1, view 2, image encoder, and text feature, respectively.
# alpha_1, alpha_2 : Learnable temperatures for sharpening the prediction.
# norm, batch_size, CE: Normalization, batch size, and cross-entropy loss, respectively.
# data_num, known_cls_num: Total number of images in a dataset and the number of known classes,
      respectively.
# count_in, count_out: Used for collecting pseudo-labels for each image. Initialized with -1s.
# omega, gamma: A balancing weight and a smoothness factor, respectively.
# delta_in, delta_out: Thresholds for IDs and OODs, respectively.

for (img_1, img_2, idx) in train_loader:

    img_feat_1, img_feat_2 = img_encoder(img_1), img_encoder(img_2)
    cos_sim_1 = alpha_1 * norm(img_feat_1, dim=1) @ norm(txt_feat, dim=1).T
    cos_sim_2 = alpha_1 * norm(img_feat_2, dim=1) @ norm(txt_feat, dim=1).T
    max_prob_1, max_idx_1 = max(softmax(cos_sim_1, dim=1), dim=1)
    max_prob_2, max_idx_2 = max(softmax(cos_sim_2, dim=1), dim=1)

    count_in, count_out, thres_in, thres_out = classwise_threshold(count_in, count_out, beta_in,
        beta_out)
    mask_in_1, mask_in_2 = max_prob_1.ge(thres_in[max_idx_1]), max_prob_2.ge(thres_in[max_idx_2])
    mask_out_1, mask_out_2 = (1-max_prob_1).ge(thres_out[max_idx_1]), (1-max_prob_2).ge(thres_out[
        max_idx_2])

    loss_in = (CE(max_prob_2,max_idx_1) *mask_in_1 + CE(max_prob_1,max_idx_2) * mask_in_2) / 2.0
    loss_out = (CE((1-max_prob_2),max_idx_1) * mask_out_1 + CE((1-max_prob_1),max_idx_2) * mask_out_2)
        / 2.0
    loss_cont = contrastive_loss(img_feat_1, img_feat_2, batch_size)

    loss = loss_in + loss_out + omega * loss_cont
    loss.backward()
    update(img_encoder.parameters())

    count_in_temp, count_out_temp = ones(data_num))*(-1), ones(data_num)*(-1)
    idx_in_1, idx_in_2 = max_prob_1.ge(delta_in), max_prob_2.ge(delta_in)
    idx_out_1, idx_out_2 = (1-max_prob_1).ge(delta_out), (1-max_prob_2).ge(delta_out)
    count_in_temp[idx[idx_in_1]], count_in_temp[idx[idx_in_2]] = max_idx_1[idx_in_1], max_idx_2[
        idx_in_2]
    count_out_temp[idx[idx_out_1]], count_out_temp[idx[idx_out_2]] = max_idx_1[idx_out_1], max_idx_2[
        idx_out_2]
    count_in_temp, count_out_temp = Counter(count_in_temp), Counter(count_out_temp)

    momentum = (batch_size*2 / data_num)
    for i in range(known_cls_num):
        count_in[i] = count_in[i]* (1-momentum) + count_in_temp[i]
        count_out[i] = count_out[i]* (1-momentum) + count_out_temp[i]

def classwise_threshold(count_in, count_out, beta_in, beta_out):

    for (img_1, img_2, idx) in train_loader:
        img = cat([img_1,img_2], dim=0)
        img_feat = img_encoder(img)
        cos_sim = alpha_1 * norm(img_feat, dim=1) @ norm(txt_feat, dim=1).T
        max_prob, max_idx = max(softmax(cos_sim, dim=1), dim=1)
        idx_in, idx_out = max_prob.ge(delta_in), (1-max_prob).ge(delta_out)
        count_in[idx[idx_in]], count_out[idx[idx_out]] = max_idx[idx_in], max_idx[idx_out]

    count_in, count_out = Counter(count_in), Counter(count_out)
    max_in, max_out = max(count_in.values()), max(count_out.values())

    for i in range(known_cls_num):
        if i in count_in:
            beta_in = (count_in[i] + gamma*max_in) / (1+gamma)*max_in
        if i in count_out:
            beta_out = (count_out[i] + gamma*max_out) / (1+gamma)*max_out

    return count_in, count_out, beta_in*delta_in, beta_out*delta_out

def contrastive_loss(feat_1, feat_2, batch_size):

    feat_1, feat_2 = norm(feat_1, dim=1), norm(feat_2, dim=1)
    label = arange(batch_size)
    mask = eye(batch_size) * 1e9
    matrix = feat_1 @ feat_2.T
    matrix1 = feat_1 @ feat_1.T - mask
    matrix2 = feat_2 @ feat_2.T - mask
    matrix1, matrix2 = cat([matrix, matrix1], dim=0), cat([matrix.T, matrix2], dim=0)
    loss = (CE(matrix1 / alpha_2, label) + CE(matrix2 / alpha_2, label)) / 2.0
    return loss
```

---

