# OpenReview forum: "Unsupervised Open-Set Task Adaptation Using a Vision-Language Foundation Model"
_ICLR.cc/2024/Conference — ICLR 2024 Conference Withdrawn Submission_

### Official Review · Reviewer_uTep · 2023-10-14

**Soundness:** 2 fair
**Presentation:** 3 good
**Contribution:** 3 good
**Rating:** 5
**Confidence:** 5

**Summary:**

This paper proposes to address open-set domain adaptation task using CLIP models. The main contribution of this paper is a method to enchance the classification performance of a zero-shot CLIP model. Technically, this work proposes two self-training losses with gradual adajustment of class-wise thresholds, reguralized by a contrastive loss. The conduct experiments on several open-set domain adaptation benchmarks and show improvements to the zero-shot method.

**Strengths:**

1.The proposed gradual adjustment of class-wise thresholds and the self-training losses for both In and OOD samples are interesting to see.

2. Experimetal results on several OSDA benchmarks are impressive.

3. The presentaion of the paper is clear to read.

**Weaknesses:**

1. This learning setting of universal black-box domain adaptation [1] covers the learning setting of this work, and the learnning is somehow similar with a self-training framework. Thus, this setting is not longer new anyway.

2. Hyperparameters analysis need comprehensive studies since we are curious about how they affect the results. Note that no validation set is available in this learning setting.

3. Some key questions need to address well, see "Questions".

[1] Deng, Bin, et al. "On universal black-box domain adaptation." arXiv preprint arXiv:2104.04665 (2021).

**Questions:**

1. What is the motivation of class-wise thresholds? What are the benifits when using a gradual adajustment of class-wise threshold?

2. In the inference stage, how to detech OOD samples? As there are two thresholds: $\delta_{in}$ and $\delta_{out}$.

3. When comparing to previous methods (e.g, In Table 2), I am cuious about whether they are run under the same CLIP backbone?

4. How the zero-shot method is used for the open-set task?

5. The self-training loss of In-distribution data likes the FixMatch [1] or not?

[1] Sohn, Kihyuk, et al. "Fixmatch: Simplifying semi-supervised learning with consistency and confidence." Advances in neural information processing systems 33 (2020): 596-608.

---

### Official Review · Reviewer_UHKB · 2023-10-17

**Soundness:** 3 good
**Presentation:** 2 fair
**Contribution:** 2 fair
**Rating:** 3
**Confidence:** 3

**Summary:**

Overall, this paper proposes a new method UODA to tune CLIP to better perform classification tasks in scenarios where only unlabelled images from a certain domain are available and among these images OOD images possibly exist.

**Strengths:**

1. The problem that this paper tries to solve is interesting, i.e., I agree that there is a need to explore the usage of CLIP in open and real-world scenario.
2. The results shown in the experimental section is awesome and the paper is well-written.

**Weaknesses:**

(See questions below for the details)

**Questions:**

Despite the strengths I above-mentioned w.r.t. this paper, I think the current version of this paper does not reach the acceptance bar of ICLR and below are my concerns:

1. The first concern I have w.r.t. this paper is whether or not it is suitable to call the problem that this paper is trying to solve a DA problem. From my basic understanding, if a problem is called a DA problem, no matter in which setting, there should at least be a source domain (even data from which is untouchable) and a target domain. However, in this paper, as stated in its table 1 that this setting neither uses a model trained on D_s or D_s itself, it seems to me that a source domain can be regarded as totally not exist. Thus, calling this setting a DA setting seems to be a confusing choice.

2. The more crucial concern I have is that, if I take this submission out of the domain adaption scope, I kind of question the novelty of this paper then. Specifically, without the domain adaptation setting, this paper seems to use a combination of (1) OOD detection through CLIP's softmax vector, (2) adapter structure in CLIP, and (3) unsupervised view augmentation and maximal seperation. While I agree on the usefulness of each of these techniques, it seems that all of them are quite common techniques already, especially for (3) (I also list recent works that have used (1) and (2) below in [1-2]). Thus, besides that it can be not suitable to call the proposed setting a DA setting and compare it with DA  methods, it also seems that this proposed setting in this paper can be tackled via combining existing techniques. Thus, I kind of question its novelty.

[1] CLIP-Adapter: Better Vision-Language Models with Feature Adapters

[2] Delving into Out-of-Distribution Detection with Vision-Language Representations

3. One remaining small question can be, it seems that the authors use two thresholds for ID and OOD. Then what if the ID threshold is far beyond the OOD threshold? How will the data in between of these two thresholds be handled?

---

### Official Review · Reviewer_G7av · 2023-10-29

**Soundness:** 1 poor
**Presentation:** 2 fair
**Contribution:** 1 poor
**Rating:** 3
**Confidence:** 4

**Summary:**

This paper offers an intuitive solution for open-set domain adaption task, where the pretrained CLIP model combined with a lightweight adapter is finetuned to adapt the target domain tasks. In this method, adaptive class-wise thresholds, in-domain and out-domain self-disitillation losses are adopted for the finetuning. The reported experimental results illustrate that this method outperforms other domain adaption method.

**Strengths:**

+ This method is very straightforward and easy to follow.
+ It combines self-disitillation idea into the finetuning of pretraining CLIP model.

**Weaknesses:**

+ The novelty is somewhat limited. The proposed method simply finetunes the pretrained CLIP model with self-disitillation loss, which doesn't provide enough technique improvement for ICLR.

+ The proposed method is only evaluated in small-scale datasets, Office-31, Office-Home and Visda, which is not enough for ICLR.

**Questions:**

+ The architecture of adapter module is the core part of this paper. However, its details are missing in the paper. Other hyper-parameters for model training are also missing.

+ The clarification of `W, D, A, D, A` in Tables and so on should be clearly explained.

+ Are there some errors about clarification, "Source-free OSDA employs models trained on labeled source data but use only target data during the adaptation stage." in the caption of Table 2.

---

### Official Review · Reviewer_a6QH · 2023-10-30

**Soundness:** 1 poor
**Presentation:** 1 poor
**Contribution:** 2 fair
**Rating:** 3
**Confidence:** 4

**Summary:**

This work aims to enhance the OOD detection and image classification capabilities of CLIP by only utilizing open-set unlabeled data. The proposed unsupervised open-set task adaptation (UOTA) mainly includes three components: 1) dynamic threshold adjustment for OOD detection; 2) self-training with in-distribution data and negative learning with OOD data; 3) contrastive regularization (i.e., SimCLR). Extensive experiments on Office-31, Office-Home, VisDA, and DomainNet show the effectiveness of the proposed algorithm.

**Strengths:**

**Originality**: The paper proposes a new setting, i.e., unsupervised open-set task adaptation, in which only unlabeled data from a specific origin and a pre-trained vision-language foundation model are available.

**Quality**: The paper provides a thorough experimental evaluation of UOTA on four open-set domain adaptation benchmarks. The paper also conducts ablation studies to analyze the impact of different objectives. The paper demonstrates that UOTA can largely improve CLIP's performance and achieve SOTA results.

**Clarity**: The paper also provides sufficient related settings to situate the contribution of UOTA in the context of existing literature on UDA, OSDA, SF-OSDA, etc.

**Significance**: This paper aims to address the important and challenging problem of unsupervised open-set task adaptation, which is crucial for the industrialization of ML methods but has not been studied enough.

**Weaknesses:**

**Major Issues**:

*Insufficient novelty and contribution*: The key factor of this work is to accurately detect OOD samples. OOD detection has been studied extensively, however, there is almost no discussion in this paper. For example, using the maximum softmax score to detect OOD samples is investigated in ref [a]. More related works should be discussed and compared with the proposed dynamic threshold adjustment for OOD detection [b,c,d,e,f,g,h].

*Insufficient results for experiments*:

- Although the authors state in the main text, "we update only this lightweight adapter, enabling computationally efficient training", they provide no experimental results. More importantly, there is almost no discussion about the adapter (parameter-efficient fine-tuning). And the motivation is unclear as well.

- For fair comparison, how does it compare with other settings (OSDA, SF-OSDA) using the proposed components? For example, using a source model pre-trained on ImageNet with additional training on labeled source data to perform SF-OSDA like ODAwVL.

- In Appendix D, the authors have provided more detailed results, but it is important to provide the detailed results of "Zero-shot".


**Minor Issues**:

- In Tab. 5, it can be seen that the performance of Zeor-shot ViT-L/14 is the worst. It does not make sense.

- Repeated references: Scaling up visual and vision-language representation learning with noisy text supervision.



Refs:

[a] Hendrycks et al. A Baseline for Detecting Misclassified and Out-of-Distribution Examples in Neural Networks. ICLR 2017.

[b] Liang et al. Principled detection of out-of-distribution examples in neural networks, arXiv 2017.

[c] DeVries et al. Learning confidence for out-of-distribution detection in neural networks. arXiv 2018.

[d] Shalev et al. Out-of-distribution detection using multiple semantic label representations. NeurIPS 2018.

[e] Liu et al. Energy-based out-of-distribution detection. NeurIPS 2020.

[f] Yang et al. Generalized out-of-distribution detection: A survey. arXiv 2021.

[g] Sun et al. Out-of-distribution detection with deep nearest neighbors. ICML 2022.

[h] Sun et al. React: Out-of-distribution detection with rectified activations. NeurIPS 2021.

**Questions:**

My first concern is that "unsupervised open-set task adaptation" should consist of three parts: unsupervised, open-set, and task adaptation. This work fulfills the first two parts while the last one of the task adaptation is completely insufficient, especially all experiments are conducted on open-set domain adaptation benchmarks. In my view, the authors should testify the proposed method in true task adaptation benchmarks such as the Visual Task Adaptation Benchmark (VTAB). In the current version, it should be called source-free open-set domain adaptation with CLIP.

Then, what is the advantage of the proposed setting in this work compared with SF-OSDA? Compared to a pre-trained CLIP model, a model trained on a source domain is more accessible and cheap.

At last, a small question, from the t-SNE visualization in Fig. 3, we can see that UOTA completely loses the discriminability on OOD samples (red), could we think that the classification capability of CLIP on in-distribution data has been improved by weakening the generalization of CLIP?